# Near Optimal Exploration-Exploitation in Non-Communicating Markov Decision Processes

**Ronan Fruit**
Sequel Team - Inria Lille
`ronan.fruit@inria.fr`

**Matteo Pirotta**
Sequel Team - Inria Lille
`matteo.pirotta@inria.fr`

**Alessandro Lazaric**
Facebook AI Research
`lazaric@fb.com`

## Abstract

While designing the state space of an MDP, it is common to include states that are transient or not reachable by any policy (e.g., in mountain car, the product space of speed and position contains configurations that are not physically reachable). This results in weakly-communicating or multi-chain MDPs. In this paper, we introduce TUCRL, the first algorithm able to perform efficient exploration-exploitation in any finite Markov Decision Process (MDP) without requiring any form of prior knowledge. In particular, for any MDP with $S^{\mathsf{c}}$ communicating states, $A$ actions and $\Gamma^{\mathsf{c}} \leq S^{\mathsf{c}}$ possible communicating next states, we derive a $\widetilde{O}(D^{\mathsf{c}}\sqrt{\Gamma^{\mathsf{c}}S^{\mathsf{c}}AT})$ regret bound, where $D^{\mathsf{c}}$ is the diameter (i.e., the length of the longest shortest path between any two states) of the communicating part of the MDP. This is in contrast with existing optimistic algorithms (e.g., UCRL, Optimistic PSRL) that suffer linear regret in weakly-communicating MDPs, as well as posterior sampling or regularised algorithms (e.g., REGAL), which require prior knowledge on the bias span of the optimal policy to achieve sub-linear regret. We also prove that in weakly-communicating MDPs, no algorithm can ever achieve a logarithmic growth of the regret without first suffering a linear regret for a number of steps that is exponential in the parameters of the MDP. Finally, we report numerical simulations supporting our theoretical findings and showing how TUCRL overcomes the limitations of the state-of-the-art.

## 1 Introduction

Reinforcement learning (RL) [1] studies the problem of learning in sequential decision-making problems where the dynamics of the environment is unknown, but can be learnt by performing actions and observing their outcome in an online fashion. A sample-efficient RL agent must trade off the *exploration* needed to collect information about the environment, and the *exploitation* of the experience gathered so far to gain as much reward as possible. In this paper, we focus on the regret framework in *infinite-horizon average-reward* problems [2], where the exploration-exploitation performance is evaluated by comparing the rewards accumulated by the learning agent and an optimal policy. Jaksch et al. [2] showed that it is possible to efficiently solve the exploration-exploitation dilemma using the *optimism in face of uncertainty* (OFU) principle. OFU methods build confidence intervals on the dynamics and reward (i.e., construct a set of plausible MDPs), and execute the optimal policy of the "best" MDP in the confidence region [e.g., 2, 3, 4, 5, 6]. An alternative approach is posterior sampling (PS) [7], which maintains a posterior distribution over MDPs and, at each step, samples an MDP and executes the corresponding optimal policy [e.g., 8, 9, 10, 11, 12].

**Weakly-communicating MDPs and misspecified states.** One of the main limitations of UCRL [2] and optimistic PSRL [12] is that they require the MDP to be communicating so that its diameter $D$ (i.e., the length of the longest path among all shortest paths between any pair of states) is finite. While assuming that all states are reachable may seem a reasonable assumption, it is rarely verified *in practice*. In fact, it requires a designer to carefully define a state space $\mathcal{S}$ that contains all reachable

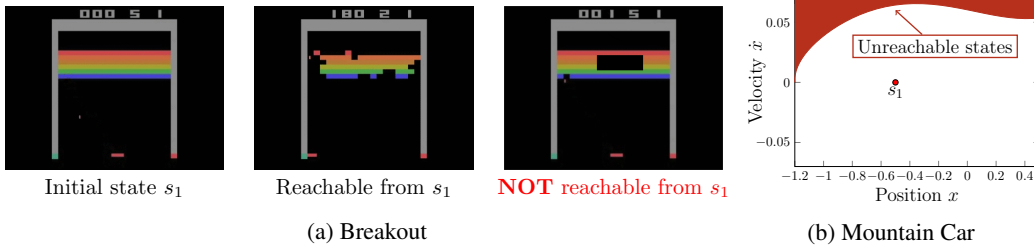

|  | (a) Breakout | | (b) Mountain Car |

Figure 1: Examples of non-communicating domains. Fig. b represents a phase plane plot of the Mountain car domain $(x, \dot{x}) \in [-1.2, 0.6] \times [-0.07, 0.07]$. The initial state is $(-0.5, 0)$ and the red area corresponds to non-reachable states from the initial state. Other non-reachable states may exist. Fig. a shows the initial state, one reachable state (*middle*) and an unreachable one (*right*).

states (otherwise it may not be possible to learn the optimal policy), but it excludes unreachable states (otherwise the resulting MDP would be non-communicating). This requires a considerable amount of prior knowledge about the environment. Consider a problem where we learn from images e.g., the Atari Breakout game [13]. The state space is the set of "plausible" configurations of the brick wall, ball and paddle positions. The situation in which the wall has an hole in the middle is a valid state (e.g., as an initial state) but it cannot be observed/reached starting from a dense wall (see Fig. 1a). As such, it should be removed to obtain a "well-designed" state space. While it may be possible to design a suitable set of "reachable" states that define a communicating MDP, this is often a difficult and tedious task, sometimes even impossible. Now consider a continuous domain e.g., the Mountain Car problem [14]. The state is decribed by the position $x$ and velocity $\dot{x}$ along the $x$-axis. The state space of this domain is usually defined as the cartesian product $[-1.2, 0.6] \times [-0.07, 0.07]$. Unfortunately, this set contains configurations that are not physically reachable as shown on Fig. 1b. The *dynamics* of the system is constrained by the *evolution equations*. Therefore, the car can not go arbitrarily fast. On the leftmost position ($x = -1.2$) the speed $\dot{x}$ cannot exceed $0$ due to the fact that such position can be reached only with velocity $\dot{x} \leq 0$. To have a higher velocity, the car would need to acquire momentum from further left (i.e., $x < -1.2$) which is impossible by design ($-1.2$ is the left-boundary of the position domain). The maximal speed reachable for $x > -1.2$ can be attained by applying the maximum acceleration at any time step starting from the state $(x, \dot{x}) = (-1.2, 0)$. This identifies the curve reported in the Fig. 1b which denotes the boundary of the unreachable region. Note that other states may not be reachable. Whenever the state space is *misspecified* or the MDP is weakly communicating (i.e., $D = +\infty$), OFU-based algorithms (e.g.,UCRL) optimistically attribute large reward and non-zero probability to reach states that have never been observed, and thus they tend to repeatedly attempt to *explore* unreachable states. This results in poor performance and linear regret. A first attempt to overcome this major limitation is REGAL.C [3] (Fruit et al. [6] recently proposed SCAL, an implementable efficient version of REGAL.C), which requires prior knowledge of an upper-bound $H$ to the span (i.e., range) of the optimal bias function $h^*$. The optimism of UCRL is then "constrained" to policies whose bias has span smaller than $H$. This implicitly "removes" non-reachable states, whose large optimistic reward would cause the span to become too large. Unfortunately, an accurate knowledge of the bias span may not be easier to obtain than designing a well-specified state space. Bartlett and Tewari [3] proposed an alternative algorithm – REGAL.D– that leverages on the *doubling trick* [15] to avoid any prior knowledge on the span. Nonetheless, we recently noticed a major flaw in the proof of [3, Theorem 3] that questions the validity of the algorithm (see App. A for further details). PS-based algorithms also suffer from similar issues.[1] To the best of our knowledge, the only regret guarantees available in the literature for this setting are [17, 18, 19]. However, the counter-example of Osband and Roy [20] seems to invalidate the result of Abbasi-Yadkori and Szepesvári [17]. On the other hand, Ouyang et al. [18] and Theocharous et al. [19] present PS algorithms with expected *Bayesian* regret scaling linearly with $H$, where $H$ is an upper-bound on the optimal bias spans of all the MDPs that can be drawn from the prior distribution ([18, Asm. 1] and [19, Sec. 5]). In [18, Remark 1], the authors claim that their algorithm does not require the knowledge of $H$ to derive the regret bound. However, in App. B we show on a very simple example that for most continuous prior distributions (e.g., uninformative priors like Dirichlet), it is very likely that $H = +\infty$ implying that the regret bound may not hold (similarly for [19]). As a

result, similarly to REGAL.C, the prior distribution should contain prior knowledge on the bias span to avoid poor performance.

In this paper, we present TUCRL, an algorithm designed to trade-off exploration and exploitation in weakly-communicating and multi-chain MDPs (e.g., MDPs with misspecified states) without any prior knowledge and under the only assumption that the agent starts from a state in a communicating subset of the MDP (Sec. 3). In communicating MDPs, TUCRL eventually (after a finite number of steps) performs as UCRL, thus achieving problem-dependent logarithmic regret. When the true MDP is weakly-communicating, we prove that TUCRL achieves a $\widetilde{O}(\sqrt{T})$ regret that with polynomial dependency on the MDP parameters. We also show that it is not possible to design an algorithm achieving logarithmic regret in weakly-communicating MDPs without having an exponential dependence on the MDP parameters (see Sec. 5). TUCRL is the first computationally tractable algorithm in the OFU literature that is able to adapt to the MDP nature without any prior knowledge. The theoretical findings are supported by experiments on several domains (see Sec. 4).

## 2    Preliminaries

We consider a finite *weakly-communicating* Markov decision process [21, Sec. 8.3] $M = \langle \mathcal{S}, \mathcal{A}, r, p \rangle$ with a set of states $\mathcal{S}$ and a set of actions $\mathcal{A} = \bigcup_{s \in \mathcal{S}} \mathcal{A}_s$. Each state-action pair $(s, a) \in \mathcal{S} \times \mathcal{A}_s$ is characterized by a reward distribution with mean $r(s, a)$ and support in $[0, r_{\max}]$ as well as a transition probability distribution $p(\cdot|s, a)$ over next states. In a weakly-communicating MDP, the state-space $\mathcal{S}$ can be *partitioned* into two subspaces [21, Section 8.3.1]: a *communicating* set of states (denoted $\mathcal{S}^{\text{c}}$ in the rest of the paper) with each state in $S^{\text{c}}$ accessible –with non-zero probability– from any other state in $S^{\text{c}}$ under some stationary deterministic policy, and a –possibly empty– set of states that are *transient* under all policies (denoted $\mathcal{S}^{\text{T}}$). We also denote by $S = |\mathcal{S}|$, $S^{\text{c}} = |\mathcal{S}^{\text{c}}|$ and $A = \max_{s \in \mathcal{S}} |\mathcal{A}_s|$ the number of states and actions, and by $\Gamma^{\text{c}} = \max_{s \in \mathcal{S}^c, a \in \mathcal{A}} \|p(\cdot|s, a)\|_0$ the maximum support of all transition probabilities $p(\cdot|s, a)$ with $s \in \mathcal{S}^{\text{c}}$. The sets $\mathcal{S}^{\text{c}}$ and $\mathcal{S}^{\text{T}}$ form a partition of $\mathcal{S}$ i.e., $\mathcal{S}^{\text{c}} \cap \mathcal{S}^{\text{T}} = \emptyset$ and $\mathcal{S}^{\text{c}} \cup \mathcal{S}^{\text{T}} = \mathcal{S}$. A deterministic policy $\pi : \mathcal{S} \to \mathcal{A}$ maps states to actions and it has an associated *long-term average reward* (or *gain*) and a *bias function* defined as

$$g_M^\pi(s) := \lim_{T \to \infty} \mathbb{E}\left[\frac{1}{T} \sum_{t=1}^{T} r(s_t, \pi(s_t))\right]; \quad h_M^\pi(s) := C\text{-}\lim_{T \to \infty} \mathbb{E}\left[\sum_{t=1}^{T} \left(r(s_t, \pi(s_t)) - g_M^\pi(s_t)\right)\right],$$

where the bias $h_M^\pi(s)$ measures the expected total difference between the rewards accumulated by $\pi$ starting from $s$ and the stationary reward in *Cesaro-limit*[2] (denoted $C\text{-}\lim$). Accordingly, the difference of bias values $h_M^\pi(s) - h_M^\pi(s')$ quantifies the (dis-)advantage of starting in state $s$ rather than $s'$. In the following, we drop the dependency on $M$ whenever clear from the context and denote by $sp_{\mathcal{S}}\{h^\pi\} := \max_{s \in \mathcal{S}} h^\pi(s) - \min_{s \in \mathcal{S}} h^\pi(s)$ the *span* of the bias function. In weakly communicating MDPs, any optimal policy $\pi^* \in \arg\max_\pi g^\pi(s)$ has *constant* gain, i.e., $g^{\pi^*}(s) = g^*$ for all $s \in \mathcal{S}$. Finally, we denote by $D$, resp. $D^{\text{c}}$, the diameter of $M$, resp. the diameter of the communicating part of $M$ (i.e., restricted to the set $\mathcal{S}^{\text{c}}$):

$$D := \max_{(s,s') \in \mathcal{S} \times \mathcal{S}, s \neq s'} \{\tau_M(s \to s')\}, \qquad D^{\text{c}} := \max_{(s,s') \in \mathcal{S}^{\text{c}} \times \mathcal{S}^{\text{c}}, s \neq s'} \{\tau_M(s \to s')\}, \qquad (1)$$

where $\tau_M(s \to s')$ is the expected time of the shortest path from $s$ to $s'$ in $M$.

**Learning problem.** Let $M^*$ be the true (*unknown*) weakly-communicating MDP. We consider the learning problem where $\mathcal{S}$, $\mathcal{A}$ and $r_{\max}$ are *known*, while sets $\mathcal{S}^{\text{c}}$ and $\mathcal{S}^{\text{T}}$, rewards $r$ and transition probabilities $p$ are *unknown* and need to be estimated on-line. We evaluate the performance of a learning algorithm $\mathfrak{A}$ after $T$ time steps by its cumulative *regret* $\Delta(\mathfrak{A}, T) = Tg^* - \sum_{t=1}^{T} r_t(s_t, a_t)$. Furthermore, we state the following assumption.

**Assumption 1.** *The initial state $s_1$ belongs to the communicating set of states $\mathcal{S}^c$.*

While this assumption somehow restricts the scenario we consider, it is fairly common in practice. For example, all the domains that are characterized by the presence of a resetting distribution (e.g., episodic problems) satisfy this assumption (e.g., mountain car, cart pole, Atari games, taxi, etc.).

**Multi-chain MDPs.** While we consider weakly-communicating MDPs for ease of notation, all our results extend to the more general case of multi-chain MDPs.[3] In this case, there may be multiple

communicating and transient sets of states and the optimal gain $g^*$ is different in each communicating subset. In this case we define $S^{\mathsf{C}}$ as the set of states that are accessible –with non-zero probability– from $s_1$ ($s_1$ included) under some stationary deterministic policy. $\mathcal{S}^{\mathsf{T}}$ is defined as the complement of $S^{\mathsf{C}}$ in $\mathcal{S}$ i.e., $\mathcal{S}^{\mathsf{T}} := \mathcal{S} \setminus S^{\mathsf{C}}$. With these new definitions of $S^{\mathsf{C}}$ and $\mathcal{S}^{\mathsf{T}}$, Asm. 1 needs to be reformulated as follows:

**Assumption 1 for Multi-chain MDPs.** *The initial state $s_1$ is accessible –with non-zero probability– from any other state in $\mathcal{S}^{\mathcal{C}}$ under some stationary deterministic policy. Equivalently, $\mathcal{S}^{\mathcal{C}}$ is a communicating set of states.*

Note that the states belonging to $\mathcal{S}^{\mathsf{T}}$ can either be transient or belong to other communicating subsets of the MDP disjoint from $\mathcal{S}^{\mathsf{C}}$. It does not really matter because the states in $\mathcal{S}^{\mathsf{T}}$ will never be visited by definition. As a result, the regret is still defined as before, where the learning performance is compared to the optimal gain $g^*(s_1)$ related to the communicating set of states $\mathcal{S}^{\mathsf{C}} \ni s_1$.

# 3    Truncated Upper-Confidence for Reinforcement Learning (TUCRL)

In this section we introduce Truncated Upper-Confidence for Reinforcement Learning (TUCRL), an optimistic online RL algorithm that efficiently balances exploration and exploitation to learn in non-communicating MDPs without prior knowledge (Fig. 2).

Similar to UCRL, at the beginning of each episode $k$, TUCRL constructs confidence intervals for the reward and the dynamics of the MDP. Formally, for any $(s, a) \in \mathcal{S} \times \mathcal{A}$ we define

$$B_{p,k}(s,a) = \left\{ \widetilde{p}(\cdot|s,a) \in \mathcal{C} : \ \forall s' \in \mathcal{S}, |\widetilde{p}(s'|s,a) - \widehat{p}(s'|s,a)| \leq \beta_{p,k}^{sas'} \right\}, \tag{2}$$

$$B_{r,k}(s,a) := [\widehat{r}_k(s,a) - \beta_{r,k}^{sa}, \widehat{r}_k(s,a) + \beta_{r,k}^{sa}] \cap [0, r_{\max}], \tag{3}$$

where $\mathcal{C} = \{p \in \mathbb{R}^S | \forall s', \ p(s') \geq 0 \wedge \sum_{s'} p(s') = 1\}$ is the $(S-1)$-probability simplex, while the size of the confidence intervals is constructed using the empirical Bernstein's inequality [22, 23] as

$$\beta_{r,k}^{sa} := \sqrt{\frac{14\widehat{\sigma}_{r,k}^2(s,a)b_{k,\delta}}{N_k^+(s,a)}} + \frac{\frac{49}{3}r_{\max}b_{k,\delta}}{N_k^\pm(s,a)}, \qquad \beta_{p,k}^{sas'} := \sqrt{\frac{14\widehat{\sigma}_{p,k}^2(s'|s,a)b_{k,\delta}}{N_k^+(s,a)}} + \frac{\frac{49}{3}b_{k,\delta}}{N_k^\pm(s,a)},$$

where $N_k(s,a)$ is the number of visits in $(s,a)$ before episode $k$, $N_k^+(s,a) := \max\{1, N_k(s,a)\}$, $N_k^\pm(s,a) := \max\{1, N_k(s,a)-1\}$, $\widehat{\sigma}_{r,k}^2(s,a)$ and $\widehat{\sigma}_{p,k}^2(s'|s,a)$ are the empirical variances of $r(s,a)$ and $p(s'|s,a)$ and $b_{k,\delta} = \ln(2SAt_k/\delta)$. The set of plausible MDPs associated with the confidence intervals is then $\mathcal{M}_k = \{M = (\mathcal{S}, \mathcal{A}, \widetilde{r}, \widetilde{p}) : \ \widetilde{r}(s,a) \in B_{r,k}(s,a), \ \widetilde{p}(\cdot|s,a) \in B_{p,k}(s,a)\}$. UCRL is optimistic w.r.t. the confidence intervals so that for all states $s$ that have never been visited the optimistic reward $\widetilde{r}(s,a)$ is set to $r_{\max}$, while all transitions to $s$ (i.e., $\widetilde{p}(s|\cdot, \cdot)$) are set to the largest value compatible with $B_{p,k}(\cdot, \cdot)$. Unfortunately, some of the states with $N_k(s,a) = 0$ may be actually unreachable (i.e., $s \in \mathcal{S}^{\mathsf{T}}$) and UCRL would uniformly explore the policy space with the hope that at least one policy reaches those (optimistically desirable) states. TUCRL addresses this issue by first constructing empirical estimates of $\mathcal{S}^{\mathsf{C}}$ and $\mathcal{S}^{\mathsf{T}}$ (i.e., the set of communicating and transient states in $M^*$) using the states that have been visited so far, that is $\mathcal{S}_k^{\mathsf{C}} := \{s \in \mathcal{S} \ | \ \sum_{a \in \mathcal{A}_s} N_k(s,a) > 0\} \cup \{s_{t_k}\}$ and $\mathcal{S}_k^{\mathsf{T}} := \mathcal{S} \setminus \mathcal{S}_k^{\mathsf{C}}$, where $t_k$ is the starting time of episode $k$.

In order to avoid optimistic exploration attempts to unreachable states, we could simply execute UCRL on $\mathcal{S}_k^{\mathsf{C}}$, which is guaranteed to contain only states in the communicating set (since $s_1 \in \mathcal{S}^{\mathsf{C}}$ by Asm. 1, we have that $\mathcal{S}_k^{\mathsf{C}} \subseteq \mathcal{S}^{\mathsf{C}}$). Nonetheless, this algorithm could *under-explore* state-action pairs that would allow discovering other states in $\mathcal{S}^{\mathsf{C}}$, thus getting stuck in a subset of the communicating states of the MDP and suffering linear regret. While the states in $\mathcal{S}_k^{\mathsf{C}}$ are guaranteed to be in the communicating subset, it is not possible to know whether states in $\mathcal{S}_k^{\mathsf{T}}$ are actually reachable from $\mathcal{S}_k^{\mathsf{C}}$ or not. Then TUCRL first "guesses" a lower bound on the probability of transition from states $s \in \mathcal{S}_k^{\mathsf{C}}$ to $s' \in \mathcal{S}_k^{\mathsf{T}}$ and whenever the maximum transition probability from $s$ to $s'$ compatible with the confidence intervals (i.e., $\widehat{p}_k(s'|s,a) + \beta_{p,k}^{sas'}$) is below the lower bound, it assumes that such transition is not possible. This strategy is based on the intuition that a transition either does not exist or it should have a sufficiently "big" mass. However, these transitions should be periodically reconsidered in order to avoid *under-exploration* issues. More formally, let $(\rho_t)_{t \in \mathbb{N}}$ be a non-increasing sequence to be defined later, for all $s' \in \mathcal{S}_k^{\mathsf{T}}$, $s \in \mathcal{S}_k^{\mathsf{C}}$ and $a \in \mathcal{A}_s$, the empirical mean $\widehat{p}_k(s'|s,a)$ and variance $\widehat{\sigma}_{p,k}^2(s'|s,a)$ are zero (i.e., this transition has never been observed so far), so the largest probability

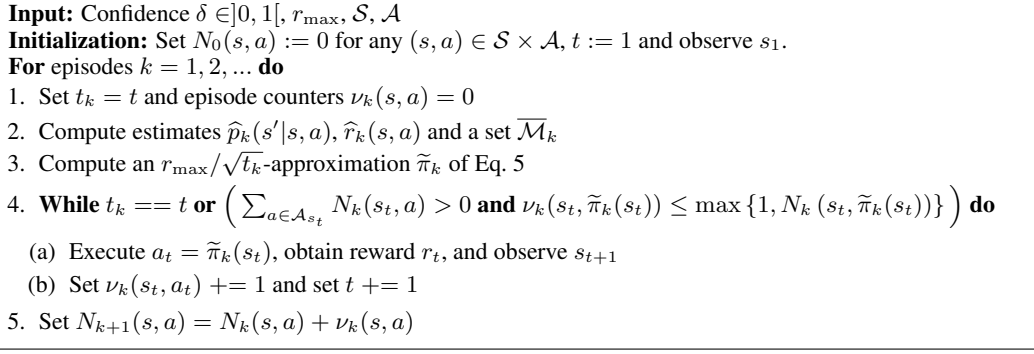

**Input:** Confidence $\delta \in ]0, 1[$, $r_{\max}$, $\mathcal{S}$, $\mathcal{A}$
**Initialization:** Set $N_0(s, a) := 0$ for any $(s, a) \in \mathcal{S} \times \mathcal{A}$, $t := 1$ and observe $s_1$.
**For** episodes $k = 1, 2, ...$ **do**
1. Set $t_k = t$ and episode counters $\nu_k(s, a) = 0$
2. Compute estimates $\widehat{p}_k(s'|s, a)$, $\widehat{r}_k(s, a)$ and a set $\overline{\mathcal{M}}_k$
3. Compute an $r_{\max}/\sqrt{t_k}$-approximation $\widetilde{\pi}_k$ of Eq. 5
4. **While** $t_k == t$ **or** $\left( \sum_{a \in \mathcal{A}_{s_t}} N_k(s_t, a) > 0 \text{ and } \nu_k(s_t, \widetilde{\pi}_k(s_t)) \leq \max\{1, N_k(s_t, \widetilde{\pi}_k(s_t))\} \right)$ **do**
   (a) Execute $a_t = \widetilde{\pi}_k(s_t)$, obtain reward $r_t$, and observe $s_{t+1}$
   (b) Set $\nu_k(s_t, a_t) += 1$ and set $t += 1$
5. Set $N_{k+1}(s, a) = N_k(s, a) + \nu_k(s, a)$

Figure 2: TUCRL algorithm.

(most optimistic) of transition from $s$ to $s'$ through any action $a$ is $\widetilde{p}_k^+(s'|s, a) = \frac{49}{3} \frac{b_{k,\delta}}{N_k^{\pm}(s,a)}$. TUCRL compares $\widetilde{p}_k^+(s'|s, a)$ to $\rho_{t_k}$ and forces all transition probabilities below the threshold to zero, while the confidence intervals of transitions to states that have already been explored (i.e., in $\mathcal{S}_k^C$) are preserved unchanged. This corresponds to constructing the alternative confidence interval

$$\overline{B}_{p,k}(s, a) = B_{p,k}(s, a) \cap \{\widetilde{p}(\cdot|s, a) \in \mathcal{C} : \forall s' \in \mathcal{S}_k^T \text{ and } \widetilde{p}_k^+(s'|s, a) < \rho_{t_k}, \widetilde{p}(s'|s, a) = 0\}. \quad (4)$$

Given $\overline{B}_{p,k}$, TUCRL (implicitly) constructs the corresponding set of plausible MDPs $\overline{\mathcal{M}}_k$ and then solves the optimistic optimization problem

$$(\widetilde{M}_k, \widetilde{\pi}_k) = \arg \max_{M \in \overline{\mathcal{M}}_k, \pi} \{g_M^\pi\}. \quad (5)$$

The resulting algorithm follows the same structure as UCRL and it is shown in Fig. 2. The episode stopping condition at line 4 is slightly modified w.r.t. UCRL. In fact, it guarantees that one action is always executed and it forces an episode to terminate as soon as a state previously in $\mathcal{S}_k^T$ is visited (i.e., $N_k(s_t, a) = 0$). This minor change guarantees that $N_{k+1}(s, a) = 0$ for all the states $s \in \mathcal{S}_k^T$ that were not reachable at the beginning of the episode. The algorithm also needs minor modifications to the extended value iteration (EVI) algorithm used to solve (5) to guarantee both efficiency and convergence. All technical details are reported in App. C.

In practice, we set $\rho_t = \frac{49 b_{t,\delta}}{3} \sqrt{\frac{SA}{t}}$, so that the *condition to remove transition* reduces to $N_k^{\pm}(s, a) > \sqrt{t_k/SA}$. This shows that only transitions from state-action pairs that have been poorly visited so far are enabled, while if the state-action pair has already been tried often and yet no transition to $s' \in \mathcal{S}_k^T$ is observed, then it is assumed that $s'$ is not reachable from $s, a$. When the number of visits in $(s, a)$ is big, the transitions to "unvisited" states should be discarded because if the transition actually exists, it is most likely extremely small and so it is worth exploring other parts of the MDP first. Symmetrically, when the number of visits in $(s, a)$ is small, the transitions to "unvisited" states should be enabled because the transitions are quite plausible and the algorithm should try to explore the outcome of taking action $a$ in $s$ and possibly reach states in $\mathcal{S}_k^T$. We denote the set of state-action pairs that are not sufficiently explored by $\mathcal{K}_k = \{(s, a) \in \mathcal{S}_k^C \times \mathcal{A} : N_k^{\pm}(s, a) \leq \sqrt{t_k/SA}\}$.

## 3.1 Analysis of TUCRL

We prove that the regret of TUCRL is bounded as follows.

**Theorem 1.** *For any weakly communicating MDP $M$, with probability at least $1 - \delta$ it holds that for any $T > 1$, the regret of TUCRL is bounded as*

$$\Delta(\text{TUCRL}, T) = O\left( r_{\max} D^C \sqrt{\Gamma^C S^C AT \ln\left(\frac{SAT}{\delta}\right)} + r_{\max} \left(D^C\right)^2 S^3 A \ln^2\left(\frac{SAT}{\delta}\right) \right).$$

The first term in the regret shows the ability of TUCRL to adapt to the communicating part of the true MDP $M^*$ by scaling with the *communicating* diameter $D^C$ and MDP parameters $S^C$ and $\Gamma^C$. The second term corresponds to the regret incurred in the early stage where the regret grows linearly.

When $M^*$ is communicating, we match the square-root term of UCRL (first term), while the second term is bigger than the one appearing in UCRL by a multiplicative factor $D^{\text{c}}S$ (ignoring logarithmic terms, see Sec. 5).

We now provide a sketch of the proof of Thm. 1 (the full proof is reported in App. D). In order to preserve readability, all following inequalities should be interpreted up to minor approximations and in high probability.

Let $\Delta_k := \sum_{s,a} \nu_k(s,a)(g^* - r(s,a))$ be the regret incurred in episode $k$, where $\nu_k(s,a)$ is the number of visits to $s, a$ in episode $k$. We decompose the regret as

$$\Delta(\text{TUCRL}, T) \lesssim \sum_{k=1}^{m} \Delta_k \cdot \mathbb{1}\{M^* \in \mathcal{M}_k\} \lesssim \sum_{k=1}^{m} \Delta_k \cdot \mathbb{1}\{t_k < C(k)\} + \sum_{k=1}^{m} \Delta_k \cdot \mathbb{1}\{t_k \geq C(k)\}$$

where $C(k) = O\left((D^{\text{c}})^2 S^3 A \ln^2(2SAt_k/\delta)\right)$ defines the length of a full exploratory phase, where the agent may suffer *linear regret*.

**Optimism.** The first technical difficulty is that whenever some transitions are disabled, the plausible set of MDPs $\overline{\mathcal{M}}_k$ may actually be *biased* and not contain the true MDP $M^*$. This requires to prove that TUCRL (i.e., the gain of the solution returned by EVI) is always optimistic despite "wrong" confidence intervals. The following lemma helps to identify the possible scenarios that TUCRL can produce (see App. D.2).[4]

**Lemma 1.** *Let episode $k$ be such that $M^* \in \mathcal{M}_k$, $\mathcal{S}_k^{\text{T}} \neq \emptyset$ and $t_k \geq C(k)$. Then, either $\mathcal{S}_k^{\text{T}} = \mathcal{S}^{\text{T}}$ (case I) or $\mathcal{K}_k \neq \emptyset$, i.e., $\exists (s,a) \in \mathcal{S}_k^{\text{C}} \times \mathcal{A}$ for which transitions to $\mathcal{S}_k^{\text{T}}$ are allowed (case II).*

This result basically excludes the case where $\mathcal{S}_k^{\text{T}} \supset \mathcal{S}^{\text{T}}$ (i.e., some states have not been reached) and yet no transition from $\mathcal{S}_k^{\text{C}}$ to them is enabled. We start noticing that when $\mathcal{S}_k^{\text{T}} = \emptyset$, the true MDP $M^* \in \mathcal{M}_k = \overline{\mathcal{M}}_k$ w.h.p. by construction of the confidence intervals. Similarly, if $\mathcal{S}_k^{\text{T}} = \mathcal{S}^{\text{T}}$ then $M^* \in \overline{\mathcal{M}}_k$ w.h.p., since TUCRL only truncates transitions that are indeed forbidden in $M^*$ itself. In both cases, we can use the same arguments in [2] to prove optimism. In *case II* the gain of any state $s' \in \mathcal{S}_k^{\text{T}}$ is set to $r_{\max}$ and, since there exists a path from $\mathcal{S}_k^{\text{C}}$ to $\mathcal{S}_k^{\text{T}}$, the gain of the solution returned by EVI is $r_{\max}$, which makes it trivially optimistic. As a result we can conclude that $\widetilde{g}_k \gtrsim g^*$ (up to the precision of EVI).

**Per-episode regret.** After bounding the optimistic reward $\widetilde{r}_k(s,a)$ w.r.t. $r(s,a)$, the only part left to bound the per-episode regret $\Delta_k$ is the term $\widetilde{\Delta}_k = \sum_{s,a} \nu_k(s,a)(\widetilde{g}_k - \widetilde{r}_k(s,a))$. Similar to UCRL, we could use the (optimistic) optimality equation and rewrite $\widetilde{\Delta}_k$ as

$$\widetilde{\Delta}_k = \sum_{s \in \mathcal{S}} \nu_k(s, \widetilde{\pi}_k(s)) \left( \sum_{s' \in \mathcal{S}} \widetilde{p}_k(s'|s, \widetilde{\pi}_k(s))\widetilde{h}_k(s') - \widetilde{h}_k(s) \right) = \nu_k' \left( \widetilde{P}_k - I \right) w_k \qquad (6)$$

where $w_k := \widetilde{h}_k - \min_{s \in \mathcal{S}}\{\widetilde{h}_k\}e$ is a shifted version of the vector $\widetilde{h}_k$ returned by EVI at episode $k$, and then proceed by bounding the difference between $\widetilde{P}_k$ and $P_k$ using standard concentration inequalities. Nonetheless, we would be left with the problem of bounding the $\ell_\infty$ norm of $w_k$ (i.e., the range of the optimistic vector $\widetilde{h}_k$) over the whole state space, i.e., $\|w_k\|_\infty = sp_{\mathcal{S}}\{\widetilde{h}_k\} = \max_{s \in \mathcal{S}} \widetilde{h}_k(s) - \min_{s \in \mathcal{S}} \widetilde{h}_k(s)$. While in communicating MDPs, it is possible to bound this quantity by the diameter of the MDP as $sp_{\mathcal{S}}\{h_k\} \leq D$ [2, Sec. 4.3], in weakly-communicating MDPs $D = +\infty$, thus making this result uninformative. As a result, we need to restrict our attention to the subset of communicating states $\mathcal{S}^{\text{C}}$, where the diameter is finite. We then split the per-step regret over states depending on whether they are explored enough or not as $\Delta_k \lesssim \sum_{s,a} \nu_k(s,a)(\widetilde{g}_k - \widetilde{r}_k(s,a))\mathbb{1}\{(s,a) \notin \mathcal{K}_k\} + r_{\max}\sum_{s,a}\nu_k(s,a)\mathbb{1}\{(s,a) \in \mathcal{K}_k\}$. We start focusing on the poorly visited state-action pairs, i.e., $(s,a) \in \mathcal{K}_k$. In this case TUCRL may suffer the maximum per-step regret $r_{\max}$ but the number of times this event happen is cumulatively "small" (App. D.4.1):

**Lemma 2.** *For any $T \geq 1$ and any sequence of states and actions $\{s_1, a_1, \ldots\ldots s_T, a_T\}$ we have:*

$$\sum_{k=1}^{m} \sum_{s,a} \nu_k(s,a) \underbrace{\mathbb{1}\{N_k^{\pm}(s,a) \leq \sqrt{t_k/SA}\}}_{(s,a)\in\mathcal{K}_k} \leq \sum_{t=1}^{T} \mathbb{1}\left\{N_{k_t}^{\pm}(s_t, a_t) \leq \sqrt{t/SA}\right\} \leq 2\left(\sqrt{S^{\text{c}}AT} + S^{\text{c}}A\right)$$

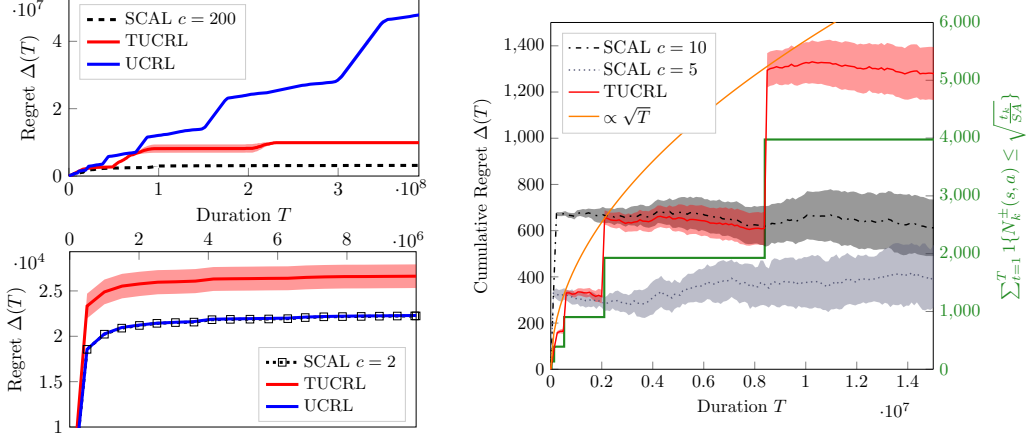

Figure 3: Cumulative regret in the taxi with misspecified states *(left-top)* and in the communicating taxi *(left-bottom)*, and in the weakly communicating three-states domain with $D = +\infty$ *(right)*. Confidence intervals $\beta_{r,k}$ and $\beta_{p,k}$ are shrunk by a factor $0.05$ and $0.01$ for the three-states domain and taxi, respectively. Results are averaged over 20 runs and 95% confidence intervals are reported.

When $(s, a) \notin \mathcal{K}_k$ (i.e., $N_k^{\pm}(s, a) > \sqrt{t_k/SA}$ holds), $\sum_{s,a} \nu_k(s, a)(\widetilde{g}_k - \widetilde{r}_k(s, a)) \cdot \mathbb{1}\{(s, a) \notin \mathcal{K}_k\}$ can be bounded as in Eq. 6 but now restricted on $\mathcal{S}_k^{\mathsf{c}}$, so that,

$$\nu_k(\widetilde{P}_k - I)\widetilde{h}_k = \sum_{s \in \mathcal{S}_k^{\mathsf{c}}} \nu_k(s, \widetilde{\pi}_k(s))\left( \sum_{s' \in \mathcal{S}_k^{\mathsf{c}}} \widetilde{p}_k(s'|s, \widetilde{\pi}_k(s))w_k(s') - w_k(s) \right).$$

Since the stopping condition guarantees that $\nu_k(s, \widetilde{\pi}_k(s)) = 0$ for all $s \in \mathcal{S}_k^{\mathsf{T}}$, we can first restrict the outer summation to states in $\mathcal{S}^{\mathsf{c}}$. Furthermore, all state-action pairs $(s, a) \notin \mathcal{K}_k$ are such that the optimistic transition probability $\widetilde{p}_k(s'|s, a)$ is forced to zero for all $s' \in \mathcal{S}_k^{\mathsf{T}}$, thus reducing the inner summation. We are then left with providing a bound for the range of $w_k$ *restricted* to the states in $\mathcal{S}_k^{\mathsf{c}}$, i.e., $sp_{\mathcal{S}_k^{\mathsf{c}}}\{w_k\} = \max_{s \in \mathcal{S}_k^{\mathsf{c}}}\{w_k\}$. We recall that EVI run on a set of plausible MDPs $\overline{\mathcal{M}}_k$ returns a function $\widetilde{h}_k$ such that $\widetilde{h}_k(s') - \widetilde{h}_k(s) \leq r_{\max} \cdot \tau_{\overline{\mathcal{M}}_k}(s \to s')$, for any pair $s, s' \in \mathcal{S}$, where $\tau_{\overline{\mathcal{M}}_k}(s \to s')$ is the expected shortest path in the extended MDP $\overline{\mathcal{M}}_k$. Furthermore, since $M^* \in \mathcal{M}_k$, for all $s, s' \in \mathcal{S}_k^{\mathsf{c}}$, $\tau_{\mathcal{M}_k}(s \to s') \leq D^{\mathsf{c}}$. Unfortunately, since $M^*$ may not belong to $\overline{\mathcal{M}}_k$, the bound on the shortest path in $\mathcal{M}_k$ (i.e., $\tau_{\mathcal{M}_k}(s \to s')$) may not directly translate into a bound for the shortest path in $\overline{\mathcal{M}}_k$, thus preventing from bounding the range of $\widetilde{h}_k$ even on the subset of states in $\mathcal{S}_k^{\mathsf{c}}$. Nonetheless, in App. E we show that a minor modification to the confidence intervals of $\overline{\mathcal{M}}_k$ makes the shortest paths between any two states $s, s' \in \mathcal{S}_k^{\mathsf{c}}$ equivalent in both sets of plausible MDPs, thus providing the bound $sp_{\mathcal{S}_k^{\mathsf{c}}}\{w_k\} \leq D^{\mathsf{c}}$. [5] The final regret in Thm. 1 is then obtained by combining all different terms.

## 4 Experiments

In this section, we present experiments to validate the theoretical findings of Sec. 3. We compare TUCRL against UCRL and SCAL.[6] We first consider the taxi problem [24] implemented in OpenAI Gym [25].[7] Even such a simple domain contains *misspecified states*, since the state space is constructed as the outer product of the taxi position, the passenger position and the destination. This leads to states that cannot be reached from any possible starting configuration (all the starting states belong to $\mathcal{S}^{\mathsf{c}}$). More precisely, out of $500$ states in $\mathcal{S}$, $100$ are non-reachable. On Fig. 3*(left)* we compare the regret of UCRL, SCAL and TUCRL when the misspecified states are present *(top)*

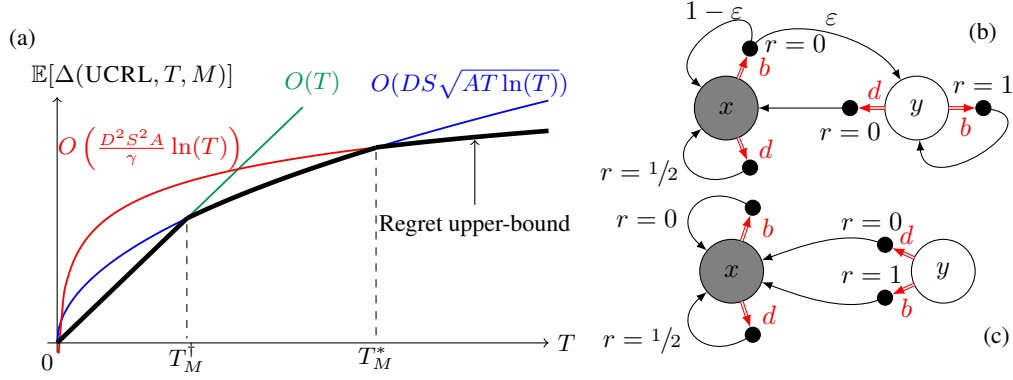

Figure 4: 4a Expected regret of UCRL (with known horizon $T$ given as input) as a function of $T$. 4b 4c Toy example illustrating the difficulty of learning non-communicating MDPs. We represent a family of possible MDPs $\mathcal{M} = (M_\varepsilon)_{\varepsilon \in [0,1]}$ where the probability $\varepsilon$ to go from $x$ to $y$ lies in $[0, 1]$.

and when they are removed *(bottom)*. In the presence of misspecified states *(top)*, the regret of UCRL clearly grows linearly with $T$ while TUCRL is able to *learn* as expected. On the other hand, when the MDP is communicating *(bottom)* TUCRL performs similarly to UCRL. The small loss in performance is most likely due to the initial exploration phase during which the confidence intervals on the transition probabilities used by UCRL (see definition of $\overline{\mathcal{M}}_k$) are tighter than those used by TUCRL (see definition of $\overline{\mathcal{M}}_k^+$). TUCRL uses a "loose" bound on the $\ell_1$-norm while UCRL uses $S$ different bounds, one for every possible next state. Finally, SCAL outperforms TUCRL by exploiting prior knowledge on the bias span.

We further study TUCRL regret in the simple three-state domain introduced in [6] (see App. H for details) with different reward distributions (uniform instead of Bernouilli). The environment is composed of only three states ($s_0$, $s_1$ and $s_2$) and one action per state, except in $s_2$ where two actions are available. As a result, the agent only has the choice between two possible policies. Fig. 3*(left)* shows the cumulative regret achieved by TUCRL and SCAL (with different upper-bounds on the bias span) when the diameter is *infinite* i.e., $\mathcal{S}^{\mathsf{C}} = \{s_0, s_2\}$ and $\mathcal{S}^{\mathsf{T}} = \{s_1\}$ (we omit UCRL, since it suffers linear regret). Both SCAL and TUCRL quickly achieve sub-linear regret as predicted by theory. However, SCAL and TUCRL seem to achieve different growth rates in regret: while SCAL appears to reach a logarithmic growth, the regret of TUCRL seems to grow as $\sqrt{T}$ with *periodic* "jumps" that are increasingly distant (in time) from each other. This can be explained by the way the algorithm works: while most of the time TUCRL is optimistic on the restricted state space $\mathcal{S}^{\mathsf{C}}$ (i.e., $\mathcal{S}_k^{\tilde{\mathsf{C}}} = \mathcal{S}^{\mathsf{C}}$), it *periodically* allows transitions to the set $\mathcal{S}^{\mathsf{T}}$ (i.e., $\mathcal{S}_k^{\mathsf{C}} = \mathcal{S}$), which is indeed not reachable. Enabling these transitions triggers aggressive *exploration* during an entire episode. The policy played is then sub-optimal creating a "jump" in the regret. At the end of this *exploratory episode*, $\mathcal{S}_k^{\tilde{\mathsf{C}}}$ will be set again to $\mathcal{S}^{\mathsf{C}}$ and the regret will stop increasing until the condition $N_k^\pm \leq \sqrt{t_k/SA}$ occurs again (the time between two consecutive exploratory episodes grows quadratically). The cumulative regret incurred during exploratory episodes can be bounded by the term plotted in green on Fig. 3*(left)*. In Lem. 2 we proved that this term is always bounded by $O(\sqrt{S^{\mathsf{C}} AT})$. Therefore, it is not surprising to observe a $\sqrt{T}$ increase of both the green and red curves. Unfortunately, the growth rate of the regret will keep increasing as $\sqrt{T}$ and will never become logarithmic unlike SCAL (or UCRL when the MDP is communicating). This is because the condition $N_k^\pm \leq \sqrt{t_k/SA}$ will always be triggered $\Theta(\sqrt{T})$ times for any $T$. In Sec. 5 we show that this is not just a drawback specific to TUCRL, but it is rather an *intrinsic limitation* of learning in weakly-communicating MDPs.

## 5 Exploration-exploitation dilemma with infinite diameter

In this section we further investigate the empirical difference between SCAL and TUCRL and prove an impossibility result *characterising* the *exploration-exploitation dilemma* when the diameter is allowed to be *infinite* and *no prior knowledge* on the optimal bias span is available.

We first recall that the expected regret $\mathbb{E}[\Delta(\text{UCRL}, M, T)]$ of UCRL (with input parameter $\delta = 1/3T$) after $T \geq 1$ time steps and for any finite MDP $M$ can be bounded in several ways:

$$\mathbb{E}[\Delta(\text{UCRL}, M, T)] \leq \begin{cases} r_{\max}T \text{ (by definition)} \\ C_1 \cdot r_{\max}D\sqrt{\Gamma SAT \ln(3T^2)} + \frac{1}{3} \text{ [2, Theorem 2]} \\ C_2 \cdot r_{\max}\frac{D^2\Gamma SA}{\gamma}\ln(T) + C_3(M) \text{ [2, Theorem 4]} \end{cases} \tag{7}$$

where $\gamma = g_M^* - \max_{s,\pi}\{g_M^\pi(s): g_M^\pi(s) < g_M^*\}$ is the gap in gain, $C_1 := 34$ and $C_2 := 34^2$ are numerical constants independent of $M$, and $C_3(M) := O(\max_{\pi:\pi(s)=a} T_\pi)$ with $T_\pi$ a measure of the "mixing time" of policy $\pi$. The three different bounds lead to three different *growth rates* for the function $T \longmapsto \mathbb{E}[\Delta(\text{UCRL}, M, T)]$ (see Fig. 4a): 1) for $T_M^\dagger \geq T \geq 0$, the expected regret is linear in $T$, 2) for $T_M^* \geq T \geq T_M^\dagger$ the expected regret grows as $\sqrt{T}$, 3) finally for $T \geq T_M^*$, the increase in regret is only logarithmic in $T$. These different *"regimes"* can be observed empirically (see [6, Fig. 5, 12]). Using (7), it is easy to show that the time it takes for UCRL to achieve sub-linear regret is at most $T_M^\dagger = \widetilde{O}(D^2\Gamma SA)$. We say that an algorithm is *efficient* when it achieves sublinear regret after a number of steps that is polynomial in the parameters of the MDP (i.e., UCRL is then *efficient*). We now show with an example that *without prior knowledge*, any *efficient* learning algorithm must satisfy $T_M^* = +\infty$ when $M$ has *infinite diameter* (i.e., it cannot achieve logarithmic regret).

**Example 1.** *We consider a family of weakly-communicating MDPs $\mathcal{M} = (M_\varepsilon)_{\varepsilon \in [0,1]}$ represented on Fig. 4(right). Every MDP instance in $\mathcal{M}$ is characterised by a specific value of $\varepsilon \in [0,1]$ which corresponds to the probability to go from $x$ to $y$. For $\varepsilon > 0$ (Fig. 4b), the optimal policy of $M_\varepsilon$ is such that $\pi^*(x) = b$ and the optimal gain is $g_\varepsilon^* = 1$ while for $\varepsilon = 0$ (Fig. 4c) the optimal policy is such that $\pi^*(x) = d$ and the optimal gain is $g_0^* = 1/2$. We assume that the learning agent knows that the true MDP $M^*$ belongs to $\mathcal{M}$ but does not know the value $\varepsilon^*$ associated to $M^* = M_{\varepsilon^*}$. We assume that all rewards are deterministic and that the agent starts in state $x$ (coloured in grey).*

**Lemma 3.** *Let $C_1, C_2, \alpha, \beta > 0$ be positive real numbers and $f$ a function defined for all $\varepsilon \in ]0, 1]$ by $f(\varepsilon) = C_1(1/\varepsilon)^\alpha$. There exists no learning algorithm $\mathfrak{A}_T$ (with known horizon $T$) satisfying both*
1. *for all $\varepsilon \in ]0, 1]$, there exists $T_\varepsilon^\dagger \leq f(\varepsilon)$ such that $\mathbb{E}[\Delta(\mathfrak{A}_T, M_\varepsilon, x, T)] < 1/6 \cdot T$ for all $T \geq T_\varepsilon^\dagger$,*
2. *and there exists $T_0^* < +\infty$ such that $\mathbb{E}[\Delta(\mathfrak{A}_T, M_0, x, T)] \leq C_2(\ln(T))^\beta$ for all $T \geq T_0^*$.*

Note that point *1* in Lem. 3 formalizes the concept of *"efficient learnability"* introduced by Sutton and Barto [26, Section 11.6] i.e., "learnable within a polynomial rather than exponential number of time steps". All the MDPs in $\mathcal{M}$ share the same number of states $S = 2 \geq \Gamma$, number of actions $A = 2$, and gap in average reward $\gamma = 1/2$. As a result, any function of $S$, $\Gamma$, $A$ and $\gamma$ will be considered as constant. For $\varepsilon > 0$, the diameter coincides with the optimal bias span of the MDP and $D = sp_\mathcal{S}\{h^*\} = 1/\varepsilon < +\infty$, while for $\varepsilon = 0$, $D = +\infty$ but $sp_\mathcal{S}\{h^*\} = 1/2$. As shown in Eq. 7 and Thm. 1, UCRL and TUCRL satisfy property *1.* of Lem. 3 with $\alpha = 2$ and $C_1 = O(S^2A)$ but do not satisfy *2.* On the other hand, SCAL satisfies *2.* with $\beta = 1$ and $C_2 = O(H^2SA/\gamma)$ (although this result is not available in the literature, it is straightforward to adapt the proof of UCRL [2, Theorem 4] to SCAL) but since [6, Theorem 12] holds only when $H \geq sp_\mathcal{S}\{h^*\}$, SCAL only satisfies *1.* for $\varepsilon \geq 1/H$ and $\varepsilon = 0$ (not for $\varepsilon \in ]0, 1/H[$). Lem. 3 proves that no algorithm can actually achieve both *1.* and *2.* As a result, since TUCRL satisfies *1.*, it cannot satisfy *2.* This matches the empirical results presented in Sec. 4 where we observed that when the diameter is infinite, the growth rates of the regret of SCAL and TUCRL were respectively logarithmic and of order $\Theta(\sqrt{T})$. An algorithm that does not satisfy *1.* could potentially satisfy *2.* but, by definition of *1.*, it would suffer linear regret for a number of steps that is more than *polynomial* in the parameters of the MDP (more precisely, $e^{D^{1/\beta}}$). This is not a very desirable property and we claim that an *efficient* learning algorithm should always prefer *finite time guarantees* (*1.*) over *asymptotic guarantees* (*2.*) when they cannot be accommodated.

## 6  Conclusion

We introduced TUCRL, an algorithm that efficiently balances exploration and exploitation in weakly-communicating and multi-chain MDPs, when the starting state $s_1$ belongs to a communicating set (Asm. 1). We showed that TUCRL achieves a square-root regret bound and that, in the general case, it is not possible to design algorithm with logarithmic regret and polynomial dependence on the MDP parameters. Several questions remain open: **1)** relaxing Asm. 1 by considering a transient initial state (i.e., $s_1 \in \mathcal{S}^T$), **2)** refining the lower bound of Jaksch et al. [2] to finally understand whether it is possible to scale with $sp_\mathcal{S}\{h^*\}$ (at least in communicating MDPs) instead of $D$ without any prior knowledge (the flaw in REGAL.D may suggest it is indeed impossible).

**Acknowledgments**

This research was supported in part by French Ministry of Higher Education and Research, Nord-Pas-de-Calais Regional Council and French National Research Agency (ANR) under project ExTra-Learn (n.ANR-14-CE24-0010-01).

## Footnotes

[1]We notice that the problem of weakly-communicating MDPs and misspecified states does not hold in the more restrictive setting of finite horizon [e.g., 8] since exploration is directly tailored to the states that are reachable *within* the known horizon, or under the assumption of the existence of a recurrent state [e.g., 16].

[2]For policies whose associated Markov chain is aperiodic, the standard limit exists.

[3]In the case of misspecified states, we implicitly define a multi-chain MDP, where each non-reachable state has a self-loop dynamics and it defines a "singleton" communicating subset.

[4]Notice that $M^* \in \mathcal{M}_k$ is true w.h.p. since $\mathcal{M}_k$ is obtained using non-truncated confidence intervals.

[5]Note that there is not a single way to modify the confidence intervals of $\overline{\mathcal{M}}_k$ to keep $sp_{\mathcal{S}_k^{\mathsf{c}}}\{w_k\}$ under control. In App. F we present an alternative modifications for which the shortest paths between any two states $s, s' \in \mathcal{S}_k^{\mathsf{c}}$ is not equal but smaller than in $\overline{\mathcal{M}}_k$ thus ensuring that $sp_{\mathcal{S}_k^{\mathsf{c}}}\{w_k\} \leq D^{\mathsf{c}}$.

[6]To the best of out knowledge, there exists no implementable algorithm to solve the optimization step of REGAL and REGAL.D.

[7]The code is available on GitHub.

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
