[Supplementary Material]

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

[8] We recently noticed that is possible to obtain a tighter relaxation that preserves the Bernstein nature of the confidence intervals (instead of resorting to $\ell_1$-norm). This version may be more efficient in practical applications. More details on this are reported in Sec. F.

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

# A   Mistake in the regret bound of REGAL.D

## A.1   Regularized optimistic RL (REGAL)

In weakly communicating MDPs, to avoid the over-optimism of UCRL, Bartlett and Tewari [3] proposed to penalise the optimism on $g^*$ by the optimal bias span $sp_{\mathcal{S}}\{h^*\}$. Formally, at each episode $k$, their algorithm –REGAL– solves the following optimization problem:

$$\widetilde{M}_k = \arg\max_{M \in \mathcal{M}_k}\{g_M^* - C_k \cdot sp_{\mathcal{S}}\{h_M^*\}\} \tag{8}$$

where $C_k \geq 0$ is a regularisation coefficient. Note that such optimization requires to first compute the optimal policy for a given MDP $M \in \mathcal{M}_k$ and then evaluate the regularized gain. Implicitly, this defines the optimistic policy $\widetilde{\pi}_k = \arg\max_{\pi \in \Pi^{\mathrm{SD}}}\{g_{\widetilde{M}_k}^\pi\}$. The term $sp_{\mathcal{S}}\{h^*\}$ can be interpreted as a measure of the *complexity* of the environment: the bigger $sp_{\mathcal{S}}\{h^*\}$, the more difficult it is to achieve the stationary reward $g^*$ by following the optimal policy. In *supervised learning*, regularisation is often used to penalise the objective function by a measure of the complexity of the model so as to avoid *overfitting*. It is thus reasonable to expect that *over-optimism* in online RL can also be avoided through regularisation.

The regret bound of REGAL holds only when $C_k$ is set to $\Theta(1/\sum_{s,a}\nu_k(s,a))$. This means that REGAL requires the knowledge of (future) visit counts $\nu_k(s,a)$ before episode $k$ begins in order to tune the regularisation coefficient $C_k$. Unfortunately, an episode stops when the number of visits in a state-action pair $(s,a) \in \mathcal{S} \times \mathcal{A}$ has doubled and it is not possible to predict the future sequence of states of a given policy for two reasons: 1) the true MDP $M^*$ is unknown and 2) what is observed is a random *sampled* trajectory (as opposed to *expected*). As a result, REGAL is not *implementable*. Bartlett and Tewari [3] proposed an alternative algorithm –REGAL.D– that leverages on the *doubling trick* to guess the length of episode $k$ (i.e., $\sum_{s,a}\nu_k(s,a)$) and proved a slightly worse regret bound than for REGAL. REGAL.D divides an episode $k$ into sub-iterations where it applies the doubling trick techniques. At each sub-iteration $j$, REGAL.D guesses that the length of the episode will be at most $2^j$ and it solves problem (8) with $C_{k,j} \propto 1/\sqrt{2^j}$. Then, it executes the optimistic policy $\widetilde{\pi}_{k,j}$ on the true MDP until the UCRL stopping condition is reached or $2^j$ steps are performed. In the first case the episode $k$ ends since the guess was correct, while, in the second case, a new sub-iteration $j+1$ is started. This implies that for any $k, j$:

$$\sum_{s,a}\nu_{k,j}(s,a) \leq 2^j, \tag{9}$$

where $\nu_{k,j}(s,a)$ denotes the number of visits to $(s,a)$ during episode $k$ and sub-iteration $j$.

## A.2   The doubling trick issue

The mistake in REGAL.D is located in the proof of the regret  [3, Theorem 3] (see Sec. 6.3). Let $\widetilde{h}_{k,j}$ denote the optimistic bias span at episode $k$ and sub-iteration $j$ induced by the doubling trick. At a high level, the mistake comes from the attempt to upper-bound the term $x \cdot \sum_{s,a}\nu_{k,j}(s,a)$ by $x \cdot 2^j$ (for a given $x$) using the fact the $\sum_{s,a}\nu_{k,j}(s,a) \leq 2^j$. Unfortunately, this is possible only under the assumption that $x \geq 0$ that does not hold in the case of REGAL.D.

Formally, while bounding $\sum_{k \in G}\Delta_k$, the authors have to deal with the term (derived by the combination of [3, Eq. 15] and [3, Lem. 11] with [3, Eq. 14]):

$$U := \sum_{k \in G}\sum_j sp_{\mathcal{S}}\left\{\widetilde{h}_{k,j}\right\}\left(c\sqrt{\sum_{s,a}\nu_{k,j}(s,a)} - C_{k,j}\sum_{s,a}\nu_{k,j}(s,a)\right)$$

where $c := 2S\sqrt{12\ln(2AT/\delta)} + \sqrt{2\ln(1/\delta)} \geq 0$ and recall that $\sum_{s,a}\nu_{k,j}(s,a)$ denotes the *actual* length of the episode $k$ at sub-iteration $j$. In the REGAL.D proof the authors directly replaced the actual length of the episode with the guessed length $2^j := \ell_{k,j}$ showing that the first term can be upper-bounded by $c \cdot \sqrt{\sum_{s,a}\nu_{k,j}(s,a)} \leq c \cdot \sqrt{2^j}$ (due to Eq. 9). Concerning the second term, they write $-C_{k,j}\sum_{s,a}\nu_{k,j}(s,a)\underset{=}{\leq} -C_{k,j}2^j$. Since $-C_{k,j} := -c/\sqrt{2^j} \leq 0$ is negative, this last inequality is

not true and the reverse inequality holds instead (using Eq. 9): $-C_{k,j} \sum_{s,a} \nu_{k,j}(s,a) \geq -C_{k,j} 2^j$. Therefore, it is not possible to guarantee that $U \leq 0$ as claimed by Bartlett and Tewari [3] (the authors probably didn't pay attention to the sign). To do this, we would need to *lower-bound* $\sum_{s,a} \nu_{k,j}(s,a)$. Unfortunately, the only lower bound with probability 1 available for that term is $\min_{s,a}\{N_k(s,a)\} + 2$. This is not big enough to cancel the term $c\sqrt{\sum_{s,a} \nu_{k,j}(s,a)}$ and $C_{k,j}$ needs to be increased. As a result, the term $sp_{\mathcal{S}}\{h^\star\} \sum_{k \in G} \sum_j C_{k,j} \sqrt{\sum_{s,a} \nu_{k,j}(s,a)}$ becomes too big and all the proof collapses.

Notice that a similar mistake is contained in the work by Maillard et al. [27] where they use a regularized approach to learn a state representation in online settings. Similarly to [3], the authors have to bound the term $\sum_{s,a} \nu_{k,j}(s,a)(g^* - \widetilde{g}_{k,j})$. By exploiting the fact that $g^* - \widetilde{g}_{k,j} \leq \alpha$ (we omit the full expression of $\alpha$ for sake of clarity) [27, Eq. 17 Sec. 5.2] the authors derived the bound $\sum_{s,a} \nu_{k,j}(s,a)(g^* - \widetilde{g}_{k,j}) \leq 2^j \cdot \alpha$ [27, Eq. 18]. The difference $g^* - \widetilde{g}_{k,j}$ might be negative in which case the result does not hold. Actually for the case in which there is no regularization $C_{k,j} = 0$, $g^* \leq \widetilde{g}_{k,j}$ which is what is used in the regret proof of UCRL. Therefore, it is very likely that the sign of $g^* - \widetilde{g}_{k,j}$ can sometimes be negative.

In conclusion, it seems unavoidable to use a *lower-bound* (and not an upper-bound) on $\sum_{s,a} \nu_{k,j}(s,a)$ to derive a correct regret bound for REGAL.D. As already mentioned, given the current stopping condition of an episode, the only reasonable lower bound is $\min_{s,a}\{N_k(s,a)\} + 2$ and it does not seem sufficient to derive a sensible regret bound. Another research direction could be to change the stopping condition. However, one of the terms in the regret bound of REGAL (and of REGAL.D) scales as $m\sqrt{T} \log_2(T)$ where $m$ is the number of episodes. The term $m$ is highly sensitive to the stopping condition and there is very little margin if we want to avoid $m\sqrt{T} \log_2(T)$ to become the leading term in the regret bound. All the efforts we put in this direction were unsuccessful. We conjecture that regularising by the optimal bias span might not allow to learn MDPs with infinite diameter.

## B  Unbounded optimal bias span with continuous Bayesian priors/posteriors

Recently, Ouyang et al. [18] and Theocharous et al. [19] proposed posterior sampling algorithms and proved bounds on the expected Bayesian regret. The regret bounds that they derive scale linearly with $H$, where $H$ is the highest optimal bias span of all the MDPs that can be drawn from the prior/posterior distribution. Formally, let $f(\theta)$ be the density function of the prior/posterior distribution over the family of MDPs $(M_\theta)$ parametrised by $\theta$. Then:

$$H := \sup_{\theta : f(\theta) > 0} \{sp_{\mathcal{S}}\{h_\theta^*\}\}.$$

In this section we present an example where $H$ is *infinite* and argue that it is probably the case for most priors/posteriors used in practice.

**Example 2** (Unbounded optimal bias span with continuous prior/posterior)**.** *Consider the example of Fig. 5. There is only one action in every state and so one optimal policy. The (unique) action that can be played in state $s_0$ loops on $s_0$ with probability $1 - \theta$ and goes to $s_1$ with probability $\theta$. The reward associated to this action is 0. Symmetrically, the (unique) action that can be played in state $s_1$ loops on $s_1$ with probability $1 - \theta$ and goes to $s_1$ with probability $\theta$. The reward associated to this action is 1. This MDP is characterised by the parameter $\theta$ and we denote it by $M_\theta$. For any $\theta \in [0, 1]$, we denote by $g_\theta^*$ (resp. $h_\theta^*$) the optimal gain (resp. bias) of $M_\theta$. Observe that when $\theta > 0$, $M_\theta$ is ergodic*

Figure 5: Toy example of a parametrised MDP $M_\theta$ with a single policy (one action per state).

*and therefore the optimal gain $g_\theta^* = 1/2$ is state-independent whereas when $\theta = 0$, $M_\theta$ is multichain and the optimal gain does depend on the initial state: $g_0^*(x) = 0 < 1 = g_0^*(y)$.*

Let's assume that the prior/posterior distribution we use on $M_\theta$ is characterised by a probability density function $f$ satisfying $f(\theta) > 0$ for all $\theta > 0$ and $f(0) = 0$. Note that this assumption does not constrain the "smoothness" of $f$ e.g., $f$ can have continuous derivatives of all orders. Under this assumption, $f$ is non-zero only for ergodic MDPs. It goes without saying that for all $\theta \in [0, 1]$ (0 included), $sp_{\mathcal{S}}\{h_\theta^*\} < +\infty$ by definition (the optimal bias span is always finite). More precisely we have:

$$g_\theta^* = \begin{cases} [1/2, 1/2]^\mathsf{T} & \text{if } \theta > 0 \\ [0, 1]^\mathsf{T} & \text{if } \theta = 0 \end{cases} \quad \text{and} \quad sp_{\mathcal{S}}\{h_\theta^*\} = \begin{cases} \frac{1}{2\theta} & \text{if } \theta > 0 \\ 0 & \text{if } \theta = 0 \end{cases}$$

As a result, although $sp_{\mathcal{S}}\{h_\theta^*\}$ is always *finite*, i.e., $\forall \theta \in [0, 1]$, $sp_{\mathcal{S}}\{h_\theta^*\} < +\infty$, it is *unbounded* on the set of plausible MDPs $\theta \in ]0, 1]$ satisfying $f(\theta) > 0$, i.e.,

$$H := \sup_{\theta \in ]0,1]} \{sp_{\mathcal{S}}\{h_\theta^*\}\} = \lim_{\theta \to 0^+} \frac{1}{2\theta} = +\infty$$

Therefore, the regret bound $\widetilde{O}(HS\sqrt{AT})$ proved by Ouyang et al. [18], Theocharous et al. [19] does not hold with prior/posterior $f$ since $H = +\infty$. One might argue that the proofs in [18, 19] could be fixed by showing that $H$ is bounded with probability 1. Unfortunately, for any $C \in [0, +\infty[$, the probability $\mathbb{P}(sp_{\mathcal{S}}\{h_\theta^*\} \geq C) = \int_{\theta=0}^{\frac{1}{2C}} f(\theta)d\theta > 0$ of sampling an MDP with $sp_{\mathcal{S}}\{h_\theta^*\} \geq C$ is strictly positive. We therefore conjecture that for this specific choice of priors/posteriors, the regret proof in [18, 19] cannot be fixed without major changes and new arguments. More generally, let's imagine that we have a prior/posterior distribution $f$ satisfying:

- there exists $\theta_0$ such that $M_{\theta_0}$ has non-constant gain i.e., $sp_{\mathcal{S}}\{g_{\theta_0}^*\} > 0$,
- there exists an open neighbourhood of $\theta_0$ denoted $\Theta_0$ such that $\forall \theta \in \Theta_0$, $M_\theta$ has constant gain (e.g., $M_\theta$ is weakly-communicating) and $f(\theta) > 0$.

In this case we will face the same problem as in Ex. 2 i.e.,

$$\sup_{\theta: \, f(\theta) > 0} \{sp_{\mathcal{S}}\{h_\theta^*\}\} = +\infty \quad \text{and} \quad \forall C \in [0, +\infty[, \, \mathbb{P}(sp_{\mathcal{S}}\{h_\theta^*\} \geq C) > 0$$

When the set of plausible MDPs is *finite*, this problem cannot occur. But most priors/posteriors used in practice are *continuous* distributions. For instance, a Dirichlet distribution will most likely satisfy the above assumptions.

## C  Algorithmic Details

For technical reasons (see App. E), we consider a slight *relaxation of the optimization problem* (5) in which $\overline{\mathcal{M}}_k$ is replaced by a relaxed extended MDP $\overline{\mathcal{M}}_k^+ \supseteq \overline{\mathcal{M}}_k$ defined by using $\ell_1$-norm concentration inequalities for $p(\cdot|s, a)$.[8] Let $B_{p,k}^+(s, a) = \{\widetilde{p}(\cdot|s, a) \in \mathcal{C} : \|\widetilde{p}(\cdot|s, a) - \widehat{p}(\cdot|s, a)\|_1 \leq \sum_{s'} \beta_{p,k}^{sas'}\}$ (resp. $\overline{B}_{p,k}^+$) be the relaxed confidence interval, then $\mathcal{M}_k^+$ (resp. $\overline{\mathcal{M}}_k^+$) is the corresponding (relaxed) set of plausible MDPs. This relaxed optimistic optimization problem is solved by running extended value iteration (EVI) on $\overline{\mathcal{M}}_k^+$ (up to accuracy $\epsilon_k = r_{\max}/\sqrt{t_k}$). Technically, we restrict EVI to work on the set of states $\mathcal{S}_k^{\mathrm{EVI}}$ that are optimistically reachable from the communicating set $\mathcal{S}_k^{\mathrm{C}}$. In practice, $\mathcal{S}_k^{\mathrm{EVI}} = \mathcal{S}_k^{\mathrm{C}}$ when $\mathcal{K}_k = \emptyset$ since all the transitions to $\mathcal{S}_k^{\mathrm{T}}$ are forbidden, otherwise $\mathcal{S}_k^{\mathrm{EVI}} = \mathcal{S}$. Alg. 1 shows this variation of EVI that we name *Truncated EVI*. Then, at each episode $k$, TUCRL runs TEVI with the following parameters: $(\widetilde{g}_k, \widetilde{h}_k, \widetilde{\pi}_k) = \mathrm{TEVI}(\mathbf{0}, \overline{\mathcal{M}}_k^+, \mathcal{S}_k^{\mathrm{EVI}}, \epsilon_k)$. Starting from an initial vector $v_0 = 0$, TEVI iteratively applies (on a subset $\mathcal{S}_k^{\mathrm{EVI}}$ of states) the optimal Bellman operator $\widetilde{L}_{\overline{\mathcal{M}}_k^+}$ associated to the (extended) MDP $\overline{\mathcal{M}}_k^+$ defined as

$$\forall v \in \mathbb{R}^S, \quad \widetilde{L}_{\overline{\mathcal{M}}_k^+} v(s) := \max_{a \in \mathcal{A}_s} \left\{ \max_{\widetilde{r} \in B_{r,k}(s,a)} \widetilde{r} + (\widetilde{p}^{sa})^\mathsf{T} v \right\}, \tag{10}$$

**Algorithm 1** TRUNCATED EXTENDED VALUE ITERATION (TEVI)

---

**Input:** value vector $v_0$, extended MDP $\mathcal{M}$, set of states $\overline{\mathcal{S}}$, accuracy $\epsilon$
**Output:** $g_n, v_n, \pi_n$
$n := 0$
$v_1(s) := \widetilde{L}_{\mathcal{M}} v_0(s) := \max_{a \in \mathcal{A}_s} \left\{ \max_{\widetilde{r} \in B_r(s,a)} \widetilde{r} + \max_{\widetilde{p} \in B_p(s,a)} \widetilde{p}^\mathsf{T} v_0 \right\}, \forall s \in \overline{\mathcal{S}}$ (see App. C)
**while** $\max_{s \in \overline{\mathcal{S}}} \{ v_{n+1}(s) - v_n(s) \} - \min_{s \in \overline{\mathcal{S}}} \{ v_{n+1}(s) - v_n(s) \} > \epsilon$ **do**
$\quad n := n + 1$
$\quad v_{n+1}(s) := \widetilde{L}_{\mathcal{M}} v_n(s), \forall s \in \overline{\mathcal{S}}$
**end while**
$g_n := \frac{1}{2} \left( \max_{s \in \overline{\mathcal{S}}} \{ v_{n+1}(s) - v_n(s) \} + \min_{s \in \overline{\mathcal{S}}} \{ v_{n+1}(s) - v_n(s) \} \right)$
$\pi_n(s) \in \arg \max_{a \in \mathcal{A}_s} \left\{ \max_{\widetilde{r} \in B_r(s,a)} \widetilde{r} + \max_{\widetilde{p} \in B_p(s,a)} \widetilde{p}^\mathsf{T} v_n \right\}, \forall s \in \overline{\mathcal{S}}$

---

where $\widetilde{p}^{sa} = \arg \max_{\widetilde{p} \in \overline{B}^+_{p,k}(s,a)} \{ \widetilde{p}^T v \}$ can be solved using [2, Fig. 2], except for $(s,a) \notin \mathcal{K}_k$ for which we force $\widetilde{p}^{sa}(s') := 0$ for any $s' \in \mathcal{S}_k^\mathsf{T}$ (see Alg. 2). If TEVI is stopped when $sp_{\mathcal{S}_k^{\text{EVI}}} \{ v_{n+1} - v_n \} \leq \epsilon_k$ and the true MDP is sufficiently explored, then the greedy policy $\widetilde{\pi}_k := \pi_n$ w.r.t. $v_n$ is $\epsilon_k$-optimistic, i.e., $\widetilde{g}_k := g_n \geq g^*_{M^*} - \epsilon_k$ (see Sec. 3.1 for details). The policy $\widetilde{\pi}_k$ is then executed until the number of visits to a state-action pair is doubled or a new state is "discovered" (i.e., $s_t \in \mathcal{S}_{k_t}^\mathsf{T}$). Note that the condition $sp_{\mathcal{S}_k^{\text{EVI}}} \{ v_{n+1} - v_n \} \leq \epsilon_k$ is always met after a *finite* number of steps since the extended MDP $\overline{\mathcal{M}}_k^+$ is communicating on the restricted state space $\mathcal{S}_k^{\text{EVI}}$. Finally, notice that when the true MDP $M^*$ is communicating, there exists an episode $\overline{k}$ s.t. for all $k \geq \overline{k}$, $\mathcal{S}_k^\mathsf{T} = \emptyset$ and TUCRL *can be reduced* to UCRL by considering $\mathcal{M}_k$ in place of $\overline{\mathcal{M}}_k^+$.

## D  Regret of TUCRL

We follow the proof structure of Jaksch et al. [2], Fruit et al. [6] and use similar notations. Nonetheless, several parts of the proof significantly differ from [2, 6]:

- in Sec. D.2 we prove that after a finite number of steps, TUCRL is *gain-optimistic* (which is not as straightforward as in the case of UCRL),
- in Sec. D.3 we show that the sums taken over the whole state space $\mathcal{S}$ that appear in the main term of the regret decomposition of UCRL can be restricted to sums over $\mathcal{S}_k^\mathsf{C}$ thanks to the new stopping condition used for episodes and the use of the condition $N_k^\pm(s,a) > \sqrt{t_k/SA}$ (see (18)),
- in Sec. D.4.1, we bound the number of time steps spent in "bad" state-action pairs $(s,a)$ satisfying $N_k^\pm(s,a) \leq \sqrt{t_k/SA}$,
- in Sec. D.4.3, we bound the number of episodes with the new stopping condition.

### D.1  Splitting into episodes

The regret of TUCRL after $T$ time steps is defined as: $\Delta(\text{TUCRL}, T) := Tg^* - \sum_{t=1}^T r_t(s_t, a_t)$. Defining $\Delta_k = \sum_{s \in \mathcal{S}, a \in \mathcal{A}} \nu_k(s,a) \left( g^* - r(s,a) \right)$ and using the same arguments as in [2, 6], it holds with probability $1 - \frac{\delta}{12T^{4/5}}$ that:

$$\Delta(\text{TUCRL}, T) \leq \sum_{k=1}^m \Delta_k + r_{\max} \sqrt{\frac{5}{2} T \ln \left( \frac{8T}{\delta} \right)} \tag{11}$$

### D.2  Episodes with $M^* \in \mathcal{M}_k$

We now assume that $M^* \in \mathcal{M}_k$. As done in App. C, let's denote by $\widetilde{g}_k$, $\widetilde{h}_k$ and $\widetilde{\pi}_k$ the outputs of TEVI$(\mathbf{0}, \mathcal{M}_k^\diamond, \mathcal{S}_k^{\text{EVI}}, \varepsilon_k)$ (see Alg. 1) where $\varepsilon_k := r_{\max}/\sqrt{t_k}$ and

$$\mathcal{S}_k^{\text{EVI}} = \begin{cases} \mathcal{S}_k^\mathsf{C} & \text{if } \mathcal{K}_k = \emptyset \\ \mathcal{S} & \text{otherwise} \end{cases}, \qquad \mathcal{M}_k^\diamond = \begin{cases} \mathcal{M}_k = \overline{\mathcal{M}}_k & \text{if } \mathcal{S}_k^\mathsf{T} = \emptyset \\ \overline{\mathcal{M}}_k^+ & \text{otherwise} \end{cases}. \tag{12}$$

TEVI returns an approximate solution of a slightly modified version of Problem 5:

$$(\widetilde{M}_k, \widetilde{\pi}_k) = \arg\max_{M \in \mathcal{M}_k^\diamond, \pi}\{g_M^\pi\}.$$

In order to bound $\Delta_k$ we first show that $\widetilde{g}_k \gtrsim g^*$ (up to $r_{\max}/\sqrt{t_k}$-accuracy). If $\mathcal{S}_k^{\mathrm{T}} = \emptyset$ then by definition $\mathcal{M}_k^\diamond = \overline{\mathcal{M}}_k = \mathcal{M}_k \ni M^*$ and so we can use the same argument as in [2, Sec. 4.3 & Thm. 7]. If $\mathcal{S}_k^{\mathrm{T}} \neq \emptyset$, the true MDP $M^*$ might not be "included" in the extended MDP $\overline{\mathcal{M}}_k^+$ considered by EVI and we cannot use the same argument. To overcome this problem we first assume that $t_k$ is big enough which allows us to prove a useful lemma (Lem. 4):

$$t_k \geq \frac{2401}{9}\left(D^{\mathtt{C}}\right)^2 SA\left(S_k^{\mathrm{T}}\ln\left(\frac{2SAt_k}{\delta}\right)\right)^2 := C(k) \tag{13}$$

where $S_k^{\mathrm{T}} := |\mathcal{S}_k^{\mathrm{T}}|$ is the cardinal of $\mathcal{S}_k^{\mathrm{T}}$.

**Lemma 4.** *Let episode $k$ be such that $M^* \in \mathcal{M}_k$, $\mathcal{S}_k^{\mathrm{T}} \neq \emptyset$ and* (13) *holds. Then,*

$$\left(\forall(s,a) \in \mathcal{S}_k^{\mathcal{C}} \times \mathcal{A}, N_k^{\pm}(s,a) > \sqrt{\frac{t_k}{SA}}\right) \implies \mathcal{S}_k^T = \mathcal{S}^T$$

*Proof.* Assume that episode $k$ is such that (13) holds and that for any state-action pair $(s,a) \in \mathcal{S}_k^{\mathcal{C}} \times \mathcal{A}$

$$N_k^{\pm}(s,a) > \sqrt{\frac{t_k}{SA}} \geq \frac{49}{3}D^{\mathtt{C}}S_k^{\mathrm{T}}\ln\left(\frac{2SAt_k}{\delta}\right)$$

Since $\mathcal{S}_k^{\mathrm{T}} \neq \emptyset$ and $M^* \in \mathcal{M}_k$, for any $(s,a,s') \in \mathcal{S}_k^{\mathcal{C}} \times \mathcal{A} \times \mathcal{S}_k^{\mathrm{T}}$

$$\underbrace{p(s'|s,a)}_{\text{transition probability in } M^*} \leq \underbrace{\widehat{p}_k(s'|s,a)}_{=0} + \beta_k^{sas'} = \underbrace{\sqrt{\frac{14\widehat{\sigma}_{p,k}^2(s'|s,a)\ln(2SAt_k/\delta)}{N_k^+(s,a)}} + \frac{49\ln(2SAt_k/\delta)}{3N_k^{\pm}(s,a)}}_{=0}$$

$$\leq \frac{49\ln(2SAt_k/\delta)}{3N_k^{\pm}(s,a)} < \frac{1}{D^{\mathtt{C}}S_k^{\mathrm{T}}}$$

where we have exploited the fact that $\widehat{p}(s'|s,a) = 0$ and $\widehat{\sigma}_{p,k}^2(s'|s,a) = 0$ for any state $s' \in \mathcal{S}_k^{\mathrm{T}}$ (remember that $N_k(s,a,s') = 0$).

We denote by $\tau_{M^*}(s \to s')$ the shortest path between any pair of states $(s,s') \in \mathcal{S} \times \mathcal{S}$ in the true MDP $M^*$. Fix an arbitrary target state $\overline{s} \in \mathcal{S}_k^{\mathrm{T}}$ and denote by $\tau(s) := \tau_{M^*}(s \to \overline{s})$ and $\tau_{\min} := \min_{s \in \mathcal{S}_k^{\mathcal{C}}}\{\tau(s)\}$. We have

$$\tau(\overline{s}) = 0$$

$$\forall s \in \mathcal{S}_k^{\mathcal{C}} \quad \tau(s) = 1 + \min_{a \in \mathcal{A}_s}\left\{\sum_{s' \in \mathcal{S}}\underbrace{p(s'|s,a)\tau(s')}_{\geq 0}\right\} \geq 1 + \min_{a \in \mathcal{A}_s}\left\{\sum_{s' \in \mathcal{S}_k^{\mathcal{C}}}p(s'|s,a)\underbrace{\tau(s')}_{\geq \tau_{\min}}\right\}$$

$$\geq 1 + \tau_{\min} \cdot \min_{a \in \mathcal{A}}\left\{\sum_{s' \in \mathcal{S}_k^{\mathcal{C}}}p(s'|s,a)\right\} = 1 + \tau_{\min} \cdot \min_{a \in \mathcal{A}}\left\{1 - \sum_{s' \in \mathcal{S}_k^{\mathrm{T}}}\underbrace{p(s'|s,a)}_{< \frac{1}{D^{\mathtt{C}}S_k^{\mathrm{T}}}}\right\}$$

$$> 1 + \tau_{\min}\left(1 - \sum_{s' \in \mathcal{S}_k^{\mathrm{T}}}\frac{1}{D^{\mathtt{C}}S_k^{\mathrm{T}}}\right) = 1 + \tau_{\min}\left(1 - \frac{1}{D^{\mathtt{C}}}\right)$$

Applying the above inequality to $\widetilde{s} \in \mathcal{S}_k^{\mathcal{C}}$ achieving $\tau(\widetilde{s}) = \tau_{\min}$ yields $\tau_{\min} > D^{\mathtt{C}}$. This implies that the shortest path in $M^*$ between any state $s \in \mathcal{S}_k^{\mathcal{C}} \subseteq \mathcal{S}^{\mathtt{C}}$ and any state in $\overline{s} \in \mathcal{S}_k^{\mathrm{T}}$ is strictly bigger than $D^{\mathtt{C}}$ but by definition $D^{\mathtt{C}}$ is the longest shortest path between any pair of states in $\mathcal{S}^{\mathtt{C}}$. Therefore, $\overline{s} \in \mathcal{S}^{\mathrm{T}}$. Since $\overline{s} \in \mathcal{S}_k^{\mathrm{T}}$ was chosen arbitrarily, then $\mathcal{S}_k^{\mathrm{T}} = \mathcal{S}^{\mathrm{T}}$. $\qquad\square$

As a consequence of Lem. 4, under the assumptions that $M^* \in \mathcal{M}_k$, $\mathcal{S}_k^{\mathsf{T}} \neq \emptyset$ and (13) holds, there are only two possible cases:

1. Either $\mathcal{S}_k^{\mathsf{T}} = \mathcal{S}^{\mathsf{T}}$,

2. or $\exists (s, a) \in \mathcal{S}_k^{\mathsf{C}} \times \mathcal{A} \; : \; N_k^{\pm}(s, a) \leq \sqrt{\frac{t_k}{SA}}$.

**Case 1:** $\mathcal{S}_k^{\mathsf{T}} = \mathcal{S}^{\mathsf{T}}$ implies that $M^* \in \overline{\mathcal{M}}_k^+$. This is because for any $(s, a, s') \in \mathcal{S}_k^{\mathsf{C}} \times \mathcal{A} \times \mathcal{S}_k^{\mathsf{T}}$ we have $p(s'|s, a) = \widetilde{p}_k(s'|s, a) = 0$ and for any $(s, a, s') \notin \mathcal{S}_k^{\mathsf{C}} \times \mathcal{A} \times \mathcal{S}_k^{\mathsf{T}}$ we have $|p(s'|s, a) - \widehat{p}_k(s'|s, a)| \leq \beta_{p,k}^{sas'}$ and so $p(\cdot|s, a) \in \overline{B}_{p,k}^+(s, a)$. Since $M^* \in \overline{\mathcal{M}}_k^+$, we can use the same argument as Jaksch et al. [2, Sec. 4.3 & Theorem 7] to prove $\widetilde{g}_k \geq g^* - \frac{r_{\max}}{\sqrt{t_k}}$.

**Case 2:** For any $(s, a) \in \mathcal{S}_k^{\mathsf{T}} \times \mathcal{A}$, $\overline{B}_{p,k}^+(s, a) = \mathcal{C}$ is the $(S-1)$-simplex denoting the maximal uncertainty about the transition probabilities, and $B_{r,k}(s, a) = [0, r_{\max}]$. We will now construct an MDP $M' \in \overline{\mathcal{M}}_k^+$ with optimal gain $r_{\max}$. For all $(s, a) \in \mathcal{S}_k^{\mathsf{T}} \times \mathcal{A}$, we set the transitions to $p_{M'}(s|s, a) = 1$ and rewards to $r_{M'}(s, a) = r_{\max}$. Let $(\overline{s}, \overline{a}) \in \mathcal{S}_k^{\mathsf{C}} \times \mathcal{A}$ such that $N_k^{\pm}(\overline{s}, \overline{a}) \leq \sqrt{\frac{t_k}{SA}}$ (which exists by assumption). We set $p_{M'}(s'|\overline{s}, \overline{a}) > 0$ for all $s' \in \mathcal{S}_k^{\mathsf{T}}$. This is possible because by definition of $\overline{\mathcal{M}}_k^+$, the support of $p(\cdot|\overline{s}, \overline{a})$ is not restricted to $\mathcal{S}_k^{\mathsf{C}}$. Finally, for all state-action pairs $(s, a) \in \mathcal{S}_k^{\mathsf{C}} \times \mathcal{A}$, we set $p_{M'}(\overline{s}|s, a) > 0$. This is possible because by definition of $\overline{\mathcal{M}}_k^+$, the support of $p(\cdot|s, a)$ is only restricted to $\mathcal{S}_k^{\mathsf{C}}$ and $\overline{s} \in \mathcal{S}_k^{\mathsf{C}}$. In $M'$, for all policies, all states in $\mathcal{S}_k^{\mathsf{T}}$ are absorbing states (i.e., loop on themselves with probability 1) with maximal reward $r_{\max}$ and all other states $s \in \mathcal{S}_k^{\mathsf{C}}$ are transient. The optimal gain of $M'$ is thus $r_{\max}$ and since $M' \in \overline{\mathcal{M}}_k^+$ we conclude that $\widetilde{g}_k \geq r_{\max} - \frac{r_{\max}}{\sqrt{t_k}} \geq g^* - \frac{r_{\max}}{\sqrt{t_k}}$.

In conclusion, TEVI is always returning an *optimistic* policy when the assumptions of Lem. 4 hold. The regret $\Delta_k$ accumulated in episode $k$ can thus be upper-bounded as:

$$\Delta_k = \sum_{s,a} \nu_k(s, a)(g^* - r(s, a)) = \sum_{s,a} \nu_k(s, a)(\underbrace{g^*}_{\leq \widetilde{g}_k + \frac{r_{\max}}{\sqrt{t_k}}} - \widetilde{r}_k(s, a)) + \sum_{s,a} \nu_k(s, a)(\widetilde{r}_k(s, a) - r(s, a))$$

$$\leq \underbrace{\sum_{s,a} \nu_k(s, a)(\widetilde{g}_k - \widetilde{r}_k(s, a))}_{:= \widetilde{\Delta}_k} + \sum_{s,a} \nu_k(s, a)(\widetilde{r}_k(s, a) - r(s, a)) + r_{\max} \sum_{s,a} \frac{\nu_k(s, a)}{\sqrt{t_k}}$$

To bound the difference between the optimistic reward $\widetilde{r}_k$ and the true reward $r$ we introduce the estimated reward $\widehat{r}_k$:

$$\forall s, a \in \mathcal{S} \times \mathcal{A}, \; \widetilde{r}_k(s, a) - r(s, a) = \underbrace{\widetilde{r}_k(s, a) - \widehat{r}_k(s, a)}_{\leq \beta_{r,k}^{sa} \text{ by construction}} + \underbrace{\widehat{r}_k(s, a) - r(s, a)}_{\leq \beta_{r,k}^{sa} \text{ since } M \in \mathcal{M}_k} \leq 2\beta_{r,k}^{sa}$$

and so in conclusion:

$$\Delta_k \leq \widetilde{\Delta}_k + \underbrace{2\sum_{s,a} \nu_k(s, a)\beta_{r,k}^{sa} + r_{\max} \sum_{s,a} \frac{\nu_k(s, a)}{\sqrt{t_k}}}_{:= U_k^1} \tag{14}$$

### D.3   Bounding $\widetilde{\Delta}_k$

The goal of this section is to bound the term $\widetilde{\Delta}_k := \sum_{s,a} \nu_k(s, a)(\widetilde{g}_k - \widetilde{r}_k(s, a))$. We start by discarding the state-action pairs $(s, a) \in \mathcal{K}_k$ that have been poorly visited so far:

$$\widetilde{\Delta}_k = \sum_{s,a} \nu_k(s, a)(\widetilde{g}_k - \widetilde{r}_k(s, a)) \underbrace{\mathbb{1}\{(s, a) \notin \mathcal{K}_k\}}_{:= \mathbb{1}_k(s,a)} + \sum_{s,a} \nu_k(s, a) \underbrace{(\widetilde{g}_k - \widetilde{r}_k(s, a))}_{\leq r_{\max}} \mathbb{1}\{(s, a) \in \mathcal{K}_k\}$$

$$\leq \underbrace{\sum_{s,a} \nu_k(s, a)(\widetilde{g}_k - \widetilde{r}_k(s, a))\mathbb{1}_k(s, a)}_{:= \widetilde{\Delta}_k'} + r_{\max} \sum_{s,a} \nu_k(s, a)\mathbb{1}\{(s, a) \in \mathcal{K}_k\} \tag{15}$$

We will now bound the term $\widetilde{\Delta}'_k = \sum_s \nu_k(s, \widetilde{\pi}_k(s))(\widetilde{g}_k - \widetilde{r}_k(s, \widetilde{\pi}_k(s))) \mathbb{1}_k(s, \widetilde{\pi}_k(s))$. We recall that the policy $\widetilde{\pi}_k$ is obtained by executing $\text{TEVI}(0, \mathcal{M}_k^\diamond, \mathcal{S}_k^{\text{EVI}}, \varepsilon_k)$ (see Alg. 1) where $\varepsilon_k := r_{\max}/\sqrt{t_k}$ and $\mathcal{S}_k^{\text{EVI}}$ and $\mathcal{M}_k^\diamond$ are defined in (12). In all possible cases for both $\mathcal{S}_k^{\text{EVI}}$ and $\mathcal{M}_k^\diamond$, this amounts to applying value iteration to a *communicating* MDP with finite state space $\mathcal{S}_k^{\text{EVI}}$ and *compact* action space. By [21, Thm. 8.5.6], since the convergence criterion of value iteration is met we have:

$$\forall s \in \mathcal{S}_k^{\text{EVI}}, \quad \left| \widetilde{h}_k(s) + \widetilde{g}_k - \widetilde{r}_k(s, \widetilde{\pi}_k(s)) - \sum_{s' \in \mathcal{S}} \widetilde{p}_k(s'|s, \widetilde{\pi}_k(s)) \widetilde{h}_k(s') \right| \leq \frac{r_{\max}}{\sqrt{t_k}} \qquad (16)$$

For all $s \notin \mathcal{S}_k^{\mathsf{c}}$, $\nu_k(s, \widetilde{\pi}_k(s)) = 0$ due to the stopping condition of episode $k$. Therefore we can plug (16) in $\widetilde{\Delta}'_k$ and derive an upper bound restricted to the set $\mathcal{S}_k^{\mathsf{c}} \subseteq \mathcal{S}_k^{\text{EVI}}$. Before to do that, we further decompose $\widetilde{\Delta}'_k$ as:

$$\begin{aligned}
\widetilde{\Delta}'_k &\leq \sum_s \nu_k(s, \widetilde{\pi}_k(s)) \left( \sum_{s' \in \mathcal{S}} \widetilde{p}_k(s'|s, \widetilde{\pi}_k(s)) \widetilde{h}_k(s') - \widetilde{h}_k(s) + \frac{r_{\max}}{\sqrt{t_k}} \right) \mathbb{1}_k(s, \widetilde{\pi}_k(s)) \\
&= \nu'_k \left( \widetilde{P}_k - I \right) \widetilde{h}_k + r_{\max} \sum_{s,a} \frac{\nu_k(s, a)}{\sqrt{t_k}} \mathbb{1}_k(s, a)
\end{aligned} \qquad (17)$$

where $\nu'_k = (\nu_k(s, \widetilde{\pi}_k(s)) \mathbb{1}_k(s, \widetilde{\pi}_k(s)))_{s \in \mathcal{S}}$ is the vector of visit counts for each state and the corresponding action chosen by $\widetilde{\pi}_k$ multiplied by the indicator function $\mathbb{1}_k$, $\widetilde{P}_k = (\widetilde{P}_k(s'|s, \widetilde{\pi}_k(s)))_{s,s' \in \mathcal{S}}$ is transition matrix associated to $\widetilde{\pi}_k$ in $\overline{\mathcal{M}}_k^+$ and $I$ is the identity matrix. We now focus on the term $\nu'_k(\widetilde{P}_k - I)\widetilde{h}_k$. Since the rows of $\widetilde{P}_k$ sum to 1, $\forall \lambda \in \mathbb{R}$, $(\widetilde{P}_k - I)\widetilde{h}_k = (\widetilde{P}_k - I)(\widetilde{h}_k + \lambda e)$ where $e = (1, \ldots 1)^{\mathsf{T}}$ is the vector of all ones. Let's take $\lambda := -\min_{s \in \mathcal{S}_k^{\mathsf{c}}}\{\widetilde{h}_k(s)\}$ and define $w_k := \widetilde{h}_k + \lambda e$ so that for all $s \in \mathcal{S}_k^{\mathsf{c}}$, $w_k(s) \geq 0$ and $\min_{s \in \mathcal{S}_k^{\mathsf{c}}}\{w_k(s)\} = 0$. We have:

$$\nu'_k(\widetilde{P}_k - I)\widetilde{h}_k = \sum_{s \in \mathcal{S}} \nu_k(s, \widetilde{\pi}_k(s)) \mathbb{1}_k(s, \widetilde{\pi}_k(s)) \left( \sum_{s' \in \mathcal{S}} \widetilde{p}_k(s'|s, \widetilde{\pi}_k(s)) w_k(s') - w_k(s) \right)$$

We denote by $k_t := \sup\{k \geq 1 : t_k \leq t\}$ the current episode at time $t$. Whenever $s_t \in \mathcal{S}_{k_t}^{\mathsf{T}}$, episode $k_t$ stops before executing any action (see the stopping condition of TUCRL in Fig. 2) implying that $\forall s \in \mathcal{S}_k^{\mathsf{T}}, \nu_k(s, \widetilde{\pi}_k(s)) = 0$. Therefore we have:

$$\nu'_k(\widetilde{P}_k - I)\widetilde{h}_k = \sum_{s \in \mathcal{S}_k^{\mathsf{c}}} \nu_k(s, \widetilde{\pi}_k(s)) \mathbb{1}_k(s, \widetilde{\pi}_k(s)) \left( \sum_{s' \in \mathcal{S}} \widetilde{p}_k(s'|s, \widetilde{\pi}_k(s)) w_k(s') - w_k(s) \right)$$

For all states $s$ such that $\mathbb{1}_k(s, \widetilde{\pi}_k(s)) = 1$, i.e., satisfying $N_k^\pm(s, \widetilde{\pi}_k(s)) > \sqrt{t_k/SA}$, we force TEVI to set $\widetilde{p}_k(s'|s, \widetilde{\pi}_k(s)) = 0, \forall s' \in \mathcal{S}_k^{\mathsf{T}}$, by construction of $\overline{\mathcal{M}}_k^+$ so that:

$$\nu'_k(\widetilde{P}_k - I)\widetilde{h}_k = \sum_{s \in \mathcal{S}_k^{\mathsf{c}}} \nu_k(s, \widetilde{\pi}_k(s)) \mathbb{1}_k(s, \widetilde{\pi}_k(s)) \left( \sum_{s' \in \mathcal{S}_k^{\mathsf{c}}} \widetilde{p}_k(s'|s, \widetilde{\pi}_k(s)) w_k(s') - w_k(s) \right) \qquad (18)$$

We can now introduce $p$:

$$\sum_{s' \in \mathcal{S}_k^{\mathsf{c}}} \widetilde{p}_k(s'|s, \widetilde{\pi}_k(s)) w_k(s') - w_k(s) = \sum_{s' \in \mathcal{S}_k^{\mathsf{c}}} \widetilde{p}_k(s'|s, \widetilde{\pi}_k(s)) w_k(s') - p(s'|s, \widetilde{\pi}_k(s)) w_k(s') \qquad (19)$$

$$+ \left( \sum_{s' \in \mathcal{S}_k^{\mathsf{c}}} p(s'|s, \widetilde{\pi}_k(s)) w_k(s') - w_k(s) \right) \qquad (20)$$

By definition $\mathcal{S}_k^{\mathsf{c}} \subseteq \mathcal{S}^{\mathsf{c}}$ and using $(1, \infty)$-Hölder's inequality , the term (19) can be bounded as $(19) \leq \|\widetilde{p}_k(\cdot|s, \widetilde{\pi}_k(s)) - p(\cdot|s, \widetilde{\pi}_k(s))\|_{1, \mathcal{S}^{\mathsf{c}}} \cdot \max_{s' \in \mathcal{S}_k^{\mathsf{c}}}\{w_k(s')\}$ where for any vector $v \in \mathbb{R}^{\mathcal{S}}$, $\|v\|_{1, \mathcal{S}^{\mathsf{c}}} := \sum_{s \in \mathcal{S}^{\mathsf{c}}} |v(s)|$. Define $\overline{s} \in \arg\max_{s \in \mathcal{S}_k^{\mathsf{c}}}\{w_k(s)\}$ and $\widetilde{s} \in \arg\min_{s \in \mathcal{S}_k^{\mathsf{c}}}\{w_k(s)\}$. By definition $\overline{s}, \widetilde{s} \in \mathcal{S}_k^{\mathsf{c}}$ and $w_k(\widetilde{s}) = \min_{s \in \mathcal{S}_k^{\mathsf{c}}}\{w_k(s)\} = 0$. By Lem. 7, we know that for all $s, s' \in \mathcal{S}_k^{\mathsf{c}}$,

the difference $w_k(s') - w_k(s) = \widetilde{h}_k(s') - \widetilde{h}_k(s)$ is upper bounded by $r_{\max} \cdot \tau_{\overline{\mathcal{M}}_k^+}(s \to s')$. We also know by Lem. 6 that for all $s, s' \in \mathcal{S}_k^{\mathsf{c}}$, $\tau_{\mathcal{M}_k^+}(s \to s') = \tau_{\overline{\mathcal{M}}_k^+}(s \to s')$. Since $M^* \in \mathcal{M}_k^+$ ($M^*$ is the true MDP), we also have that for all $s, s' \in \mathcal{S}_k^{\mathsf{c}} \subseteq \mathcal{S}^{\mathsf{c}}$, $\tau_{\mathcal{M}_k^+}(s \to s') \leq \tau_{M^*}(s \to s') \leq D^{\mathsf{c}}$. In conclusion, $\forall s, s' \in \mathcal{S}_k^{\mathsf{c}}$, $w_k(s') - w_k(s) \leq r_{\max} D^{\mathsf{c}}$ and in particular $\max_{s' \in \mathcal{S}_k^{\mathsf{c}}}\{w_k(s')\} = w_k(\bar{s}) = w_k(\bar{s}) - w_k(\widetilde{s}) \leq r_{\max} D^{\mathsf{c}}$. Similarly to what we did to bound $|\widetilde{r}_k - r|$ (14), we bound the distance in $\ell_1$-norm between $\widetilde{p}_k$ and $p$ by introducing $\widehat{p}_k$:

$$\|\widetilde{p}_k - p\|_{1, \mathcal{S}^{\mathsf{c}}} \leq \|\widetilde{p}_k - \widehat{p}_k\|_{1, \mathcal{S}^{\mathsf{c}}} + \|\widehat{p}_k - p\|_{1, \mathcal{S}^{\mathsf{c}}} \leq 2 \left( \sum_{s' \in \mathcal{S}^{\mathsf{c}}} \beta_{p,k}^{s\widetilde{\pi}_k(s)s'} \right) \tag{21}$$

We now bound the contribution of the term (20). Jaksch et al. [2] decompose this term into a martingale difference sequence and a telescopic sum but due to the indicator function $\mathbb{1}_k$, in our case the sum is not telescopic anymore and an additional term appears.

$$(20) = \sum_{s \in \mathcal{S}} \nu_k(s, \widetilde{\pi}_k(s)) \mathbb{1}_k(s, \widetilde{\pi}_k(s)) \left( \sum_{s' \in \mathcal{S}_k^{\mathsf{c}}} \underbrace{p(s'|s, \widetilde{\pi}_k(s)) w_k(s')}_{\geq 0, \ \forall s' \in \mathcal{S}_k^{\mathsf{c}}} \mathbb{1}\{s \in \mathcal{S}_k^{\mathsf{c}}\} - w_k(s) \mathbb{1}\{s \in \mathcal{S}_k^{\mathsf{c}}\} \right)$$

$$\leq \sum_{s \in \mathcal{S}} \nu_k(s, \widetilde{\pi}_k(s)) \mathbb{1}_k(s, \widetilde{\pi}_k(s)) \left( \sum_{s' \in \mathcal{S}_k^{\mathsf{c}}} p(s'|s, \widetilde{\pi}_k(s)) w_k(s') - w_k(s) \mathbb{1}\{s \in \mathcal{S}_k^{\mathsf{c}}\} \right)$$

$$= \sum_{t=t_k}^{t_{k+1}-1} \left( \sum_{s' \in \mathcal{S}_k^1} p(s'|s_t, \widetilde{\pi}_k(s_t)) w_k(s') - w_k(s_t) \mathbb{1}\{s_t \in \mathcal{S}_k^{\mathsf{c}}\} \right) \mathbb{1}_k(s_t, \widetilde{\pi}_k(s_t))$$

$$= \underbrace{\sum_{t=t_k}^{t_{k+1}-1} \left( \sum_{s' \in \mathcal{S}_k^1} p(s'|s_t, \widetilde{\pi}_k(s_t)) w_k(s') - w_k(s_{t+1}) \mathbb{1}\{s_{t+1} \in \mathcal{S}_k^{\mathsf{c}}\} \right) \mathbb{1}_k(s_t, \widetilde{\pi}_k(s_t))}_{:=X_t} \tag{22}$$

$$+ \underbrace{\sum_{t=t_k}^{t_{k+1}-1} \left( w_k(s_{t+1}) \mathbb{1}\{s_{t+1} \in \mathcal{S}_k^{\mathsf{c}}\} - w_k(s_t) \mathbb{1}\{s_t \in \mathcal{S}_k^{\mathsf{c}}\} \right) \mathbb{1}_k(s_t, \widetilde{\pi}_k(s_t))}_{\text{not telescopic due to } \mathbb{1}_k!} \tag{23}$$

Define the filtration $\mathcal{F}_t = \sigma(s_1, a_1, r_1, \ldots, s_{t+1})$. Since $k_t$ is $\mathcal{F}_{t-1}$-measurable:

$$\mathbb{E}\left[ w_{k_t}(s_{t+1}) \mathbb{1}\{s_{t+1} \in \mathcal{S}_{k_t}^{\mathsf{c}}\} \mathbb{1}_{k_t}(s_t, \widetilde{\pi}_{k_t}(s_t)) | \mathcal{F}_{t-1} \right] = \underbrace{\sum_{s' \in \mathcal{S}_{k_t}^{\mathsf{c}}} p(s'|s_t, \widetilde{\pi}_{k_t}(s_t)) w_{k_t}(s') \mathbb{1}_{k_t}(s_t, \widetilde{\pi}_{k_t}(s_t))}_{\mathcal{F}_{t-1}-\text{measurable}}$$

implying $\mathbb{E}[X_t | \mathcal{F}_{t-1}] = 0$ and so $(X_t, \mathcal{F}_t)_{t \geq 1}$ is a martingale difference sequence (MDS) with $|X_t| \leq r_{\max} D^{\mathsf{c}}$. We will bound (22) in the next section (Sec. D.4) using Azuma's inequality. Using the fact that $\mathbb{1}_k(s_t, \widetilde{\pi}_k(s_t)) = \mathbb{1}\{(s_t, \widetilde{\pi}_k(s_t)) \notin \mathcal{K}_k\} = 1 - \mathbb{1}\{(s_t, \widetilde{\pi}_k(s_t)) \in \mathcal{K}_k\}$ we can make a telescopic sum appear and rewrite (23) as:

$$(23) = \underbrace{\sum_{t=t_k}^{t_{k+1}-1} w_k(s_{t+1}) \mathbb{1}\{s_{t+1} \in \mathcal{S}_k^{\mathsf{c}}\} - w_k(s_t) \mathbb{1}\{s_t \in \mathcal{S}_k^{\mathsf{c}}\}}_{= w_k(s_{t_{k+1}}) \mathbb{1}\{s_{t_{k+1}} \in \mathcal{S}_k^{\mathsf{c}}\} - w_k(s_{t_k}) \mathbb{1}\{s_{t_k} \in \mathcal{S}_k^{\mathsf{c}}\} \leq r_{\max} D^{\mathsf{c}}} \text{ (telescopic sum)}$$

$$+ \sum_{t=t_k}^{t_{k+1}-1} \underbrace{\left( w_k(s_t) \mathbb{1}\{s_t \in \mathcal{S}_k^{\mathsf{c}}\} - w_k(s_{t+1}) \mathbb{1}\{s_{t+1} \in \mathcal{S}_k^{\mathsf{c}}\} \right)}_{\leq r_{\max} D^{\mathsf{c}}} \mathbb{1}\{(s_t, \widetilde{\pi}_k(s_t)) \in \mathcal{K}_k\}$$

$$\leq r_{\max} D^{\mathsf{c}} + r_{\max} D^{\mathsf{c}} \sum_{s,a} \nu_k(s, a) \mathbb{1}\{(s_t, \widetilde{\pi}_k(s_t)) \in \mathcal{K}_k\} \tag{24}$$

By gathering (15), (17), (21), (22) and (24) we obtain the following bound for $\widetilde{\Delta}_k$:

$$
\begin{aligned}
\widetilde{\Delta}_k \leq{}& 2r_{\max}D^{\complement}\sum_{s,a}\sum_{s'\in\mathcal{S}^{\complement}}\underbrace{\mathbb{1}_k(s,a)}_{\leq 1}\underbrace{\nu_k(s,a)\beta_{p,k}^{sas'}}_{\geq 0}+\sum_{t=t_k}^{t_{k+1}-1}X_t+r_{\max}D^{\complement}\\
&+r_{\max}(D^{\complement}+1)\sum_{s,a}\nu_k(s,a)\mathbb{1}\{N_k^{\pm}(s,a)\leq\sqrt{t_k/SA}\}+r_{\max}\sum_{s,a}\underbrace{\frac{\nu_k(s,a)}{\sqrt{t_k}}}_{\geq 0}\underbrace{\mathbb{1}_k(s,a)}_{\leq 1}\\
\leq{}& 2r_{\max}D^{\complement}\sum_{s,a}\sum_{s'\in\mathcal{S}^{\complement}}\nu_k(s,a)\beta_{p,k}^{sas'}+r_{\max}(D^{\complement}+1)\sum_{s,a}\nu_k(s,a)\mathbb{1}\{(s,a)\in\mathcal{K}_k\}\\
&+\sum_{t=t_k}^{t_{k+1}-1}X_t+r_{\max}D^{\complement}+r_{\max}\sum_{s,a}\frac{\nu_k(s,a)}{\sqrt{t_k}}:=U_k^2
\end{aligned}
\tag{25}
$$

**D.4   Summing over episodes with $M^*\in\mathcal{M}_k$ and $t_k\geq C(k)$**

Denote by $\mathbb{1}(k):=\mathbb{1}\{t_k\geq C(k)\}\cdot\mathbb{1}\{M^*\in\mathcal{M}_k\}$ the indicator function taking value 1 only when both $M^*\in\mathcal{M}_k$ and $t_k\geq C(k)$. By gathering (14) and (25) we obtain:

$$
\sum_{k=1}^{m}\Delta_k\cdot\mathbb{1}(k)\leq\sum_{k=1}^{m}\underbrace{U_k^1}_{\geq 0}\cdot\underbrace{\mathbb{1}(k)}_{\leq 1}+\sum_{k=1}^{m}\underbrace{U_k^2}_{\geq 0}\cdot\underbrace{\mathbb{1}(k)}_{\leq 1}\leq\sum_{k=1}^{m}U_k^1+U_k^2
\tag{26}
$$

and so

$$
\begin{aligned}
\sum_{k=1}^{m}\Delta_k\cdot\mathbb{1}(k)\leq{}&2\sum_{k=1}^{m}\sum_{s,a}\nu_k(s,a)\left(r_{\max}D^{\complement}\sum_{s'\in\mathcal{S}^{\complement}}\beta_{p,k}^{sas'}+\beta_{r,k}^{s,a}\right)+2r_{\max}\sum_{k=1}^{m}\sum_{s,a}\frac{\nu_k(s,a)}{\sqrt{t_k}}\\
&+r_{\max}(D^{\complement}+1)\sum_{k=1}^{m}\sum_{s,a}\nu_k(s,a)\mathbb{1}\{(s,a)\in\mathcal{K}_k\}+\sum_{t=1}^{T}X_t\mathbb{1}(k_t)+r_{\max}mD^{\complement}
\end{aligned}
\tag{27}
$$

We will now upper-bound the terms appearing in (27). The main novelty of (27) compared to UCRL is the term $\sum_{k=1}^{m}\sum_{s,a}\nu_k(s,a)\mathbb{1}\{(s,a)\in\mathcal{K}_k\}$ which is not present in the proof of Jaksch et al. [2]. We will show in the next section that this term is bounded by $O(\sqrt{S^{\complement}AT})$. All the other terms are similar to those found in UCRL.

**D.4.1   Poorly visited state-action pairs**

We first notice that by definition $t_{k_t}\leq t$ where $k_t:=\sup\{k\geq 1:\ t_k\leq t\}$ is the current episode at time $t$. As a result,

$$
\mathbb{1}\{(s,a)\in\mathcal{K}_{k_t}\}:=\mathbb{1}\left\{N_{k_t}^{\pm}(s_t,a_t)\leq\sqrt{t_{k_t}/SA}\right\}\leq\mathbb{1}\left\{N_{k_t}^{\pm}(s_t,a_t)\leq\sqrt{t/SA}\right\}
$$

Instead of directly bounding $\sum_{k=1}^{m}\sum_{s,a}\nu_k(s,a)\mathbb{1}\{(s,a)\in\mathcal{K}_k\}$ we will bound the number of visits $Z_T$ in state-action pairs that have been visited less than $\sqrt{t/SA}$ times

$$
Z_T:=\sum_{t=1}^{T}\mathbb{1}\left\{N_{k_t}^{\pm}(s_t,a_t)\leq\sqrt{t/SA}\right\}\geq\sum_{k=1}^{m}\sum_{s,a}\nu_k(s,a)\mathbb{1}\{(s,a)\in\mathcal{K}_k\}
$$

Note that the quantity $N_k(s,a)$ is updated only after the end of episode $k$ and the stopping condition of episodes used by TUCRL implies that (see Fig. 2):

$$
\forall k\geq 1,\ \forall(s,a)\in\mathcal{S}\times\mathcal{A},\ \nu_k(s,a)\leq N_k^+(s,a)
\tag{28}
$$

Moreover, for all $(s,a)\notin\mathcal{S}^{\complement}\times\mathcal{A}$, $\nu_k(s,a)=0$ implying that only the states $s\in S^{\complement}$ should be considered in the above sums. Using (28), we prove the following lemma:

**Lemma 5.** *For any $T \geq 1$ and any sequence of states and actions $\{s_1, a_1, \ldots \ldots s_T, a_T\}$ we have:*

$$Z_T \leq 2\sqrt{S^c A T} + 2S^c A.$$

*Proof.* For any episode $k$ starting at time $t_k$, and for any state-action pair $(s, a)$ we recall that $N_k(s, a)$ denotes the number of visits in $(s, a)$ prior to episode $k$ ($k$ not included) and by $\nu_k(s, a)$ the number of visits in $(s, a)$ during episode $k$:

$$N_k(s, a) := \sum_{t=1}^{t_k - 1} \mathbb{1}\{(s_t, a_t) = (s, a)\} \text{ and } \nu_k(s, a) := \sum_{t=t_k}^{t_{k+1} - 1} \mathbb{1}\{(s_t, a_t) = (s, a)\}$$

and so $_k(s, a) = \sum_{i=1}^{k-1} \nu_i(s, a)$. By convention, we denote by $N_{k_T + 1}(s, a) := \sum_{t=1}^{T} \mathbb{1}\{(s_t, a_t) = (s, a)\}$ the total number of visits in $(s, a)$ after $T$ time steps ($T$ included). We first decompose $Z_T$ as:

$$Z_T := \sum_{s,a} \sum_{t=1}^{T} \mathbb{1}\Big\{ \max\{1, N_{k_t}(s, a) - 1\} \leq \sqrt{t/SA} \Big\} \cdot \mathbb{1}\Big\{(s_t, a_t) = (s, a)\Big\} = \sum_{s \in S^c} \sum_a Z_T(s, a)$$

where $Z_T(s, a) := \sum_{t=1}^{T} \mathbb{1}\Big\{ \max\{1, N_{k_t}(s, a) - 1\} \leq \sqrt{t/SA} \Big\} \cdot \mathbb{1}\Big\{(s_t, a_t) = (s, a)\Big\}$

Using the fact that for all $t \geq 1$, $t_{k_t} \leq t \leq t_{k_t + 1} - 1$ we have:

$$\forall T \geq \tau \geq 1, \ Z_\tau(s, a) = \sum_{t=1}^{\tau} \underbrace{\mathbb{1}\Big\{ \max\{1, N_{k_t}(s, a) - 1\} \leq \sqrt{t/SA} \Big\}}_{\leq 1} \cdot \underbrace{\mathbb{1}\{(s_t, a_t) = (s, a)\}}_{\geq 0}$$

$$\leq \sum_{t=1}^{\tau} \mathbb{1}\{(s_t, a_t) = (s, a)\} \leq \sum_{t=1}^{t_{k_\tau + 1} - 1} \mathbb{1}\{(s_t, a_t) = (s, a)\}$$

$$= N_{k_\tau + 1}(s, a) \tag{29}$$

Let's define $t_{s,a}$ as the last time that $Z_t(s, a)$ was incremented by 1:

$$t_{s,a} := \max \Big\{ T \geq t \geq 1 : \max\{1, N_{k_t}(s, a) - 1\} \leq \sqrt{t/SA} \text{ and } (s_t, a_t) = (s, a) \Big\}$$

$$= \min \Big\{ T \geq t \geq 1 : Z_t(s, a) = Z_T(s, a) \Big\}$$

We denote by $m_{s,a} := k_{t_{s,a}}$ the corresponding episode. By definition,

$$Z_T(s, a) = Z_{t_{s,a}}(s, a) \tag{30}$$

and

$$\max\{1, N_{m_{s,a}}(s, a) - 1\} \leq \sqrt{t_{s,a}/SA} \tag{31}$$

Using (29) with $\tau = t_{s,a}$ we obtain:

$$Z_{t_{s,a}} \leq N_{m_{s,a} + 1}(s, a) \tag{32}$$

Moreover, by definition of $N_k(s, a)$ and (28):

$$N_{m_{s,a} + 1}(s, a) = N_{m_{s,a}}(s, a) + \underbrace{\nu_{m_{s,a}}(s, a)}_{\leq N_{m_{s,a}}^+(s,a)} \leq 2 \underbrace{\max\{1, N_{m_{s,a}}(s, a)\}}_{\leq \max\{1, N_{m_{s,a}}(s,a) - 1\} + 1}$$

$$\implies N_{m_{s,a} + 1}(s, a) \leq 2 \cdot \max\{1, N_{m_{s,a}}(s, a) - 1\} + 2 \tag{33}$$

Gathering (30), (31), (32), and (33) we obtain:

$$Z_T(s, a) = Z_{t_{s,a}}(s, a) \leq \max\{1, N_{m_{s,a} + 1}(s, a) - 1\} + 1 \leq 2 \cdot \max\{1, N_{m_{s,a}}(s, a) - 1\} + 2$$

$$\leq 2\sqrt{t_{s,a}/SA} + 2$$

$$\leq 2\sqrt{T/SA} + 2$$

$$\implies Z_T = \sum_{s \in S^c} \sum_a Z_T(s, a) \leq 2\sqrt{S^c A T} + 2S^c A$$

where for the last inequality we used the fact that $S^c \leq S$ (by definition) implying $S^c/\sqrt{S} = \sqrt{S^c/S} \cdot \sqrt{S^c} \leq \sqrt{S^c}$. This concludes the proof. □

As a consequence of Lem. 5:

$$\sum_{k=1}^{m}\sum_{s,a}\nu_k(s,a)\mathbb{1}\{N_k^{\pm}(s,a) \le \sqrt{t_k/S^{\mathsf{c}}A}\} \le Z_T \le 2\sqrt{S^{\mathsf{c}}AT} + 2S^{\mathsf{c}}A \tag{34}$$

### D.4.2   Confidence bounds $\beta_{r,k}^{sa}$ and $\beta_{p,k}^{sas'}$

Since (28) holds, Lemma 19 of Jaksch et al. [2] can still be applied. Moreover, exploiting again the fact that for all $(s,a) \notin \mathcal{S}^{\mathsf{c}} \times \mathcal{A}$, $\nu_k(s,a) = 0$ we obtain

$$\sum_{k=1}^{m}\sum_{s,a}\frac{\nu_k(s,a)}{\sqrt{t_k}} \le \sum_{k=1}^{m}\sum_{s,a}\frac{\nu_k(s,a)}{\sqrt{N_k^{+}(s,a)}} \le \left(\sqrt{2}+1\right)\sqrt{S^{\mathsf{c}}AT} \tag{35}$$

and as shown in [6, Appendix F.7] (with the difference that $S$ is restricted to $S^{\mathsf{c}}$) we have:

$$\sum_{k=1}^{m}\sum_{s,a}\frac{\nu_k(s,a)}{N_k^{\pm}(s,a)} \le 6S^{\mathsf{c}}A + 2S^{\mathsf{c}}A\ln(T) \tag{36}$$

The terms $\sum_{k=1}^{m}\sum_{s,a}\nu_k(s,a)\beta_{r,k}^{sa}$ and $\sum_{k=1}^{m}\sum_{s,a,s'\in\mathcal{S}^{\mathsf{c}}}\nu_k(s,a)\beta_{p,k}^{sas'}$ can then be bounded exactly as in [6, App. F.7] with $S$ replaced by $S^{\mathsf{c}}$ (except in the logarithm).

### D.4.3   Number of episodes

The stopping condition of episodes used by TUCRL (see Fig. 2) combines the original stopping condition of UCRL with the condition $s_t \in \mathcal{S}_{k_t}^{\mathsf{T}}$. Using only inequality (28), Jaksch et al. [2, Figure 1] proved that for any any sequence $\{s_1, a_1, \ldots, s_T, a_T\}$, the number of episodes is bounded by $1 + 2SA + SA\log_2\left(\frac{T}{SA}\right)$. Since (28) also holds in our case, the total number of episodes $m$ after $T$ time steps can be bounded by the same quantity (with $S$ replaced by $S^{\mathsf{c}}$ since sates in $\mathcal{S}^{\mathsf{T}}$ will never be visited) plus the number of times the event $s_t \in \mathcal{S}_{k_t}^{\mathsf{T}}$ occurs. Since whenever $s_t \in \mathcal{S}_{k_t}^{\mathsf{T}}$ state $s_t$ is removed from $\mathcal{S}_{k_t+1}^{\mathsf{T}}$ and $s_t$ necessarily belongs to $\mathcal{S}^{\mathsf{c}}$ (by definition), this event can happen at most $S^{\mathsf{c}}$ times. By Proposition 18 in [2] we thus have:

$$m \le 1 + 2S^{\mathsf{c}}A + S^{\mathsf{c}}A\log_2\left(\frac{T}{S^{\mathsf{c}}A}\right) + S^{\mathsf{c}} \tag{37}$$

### D.4.4   Martingale Difference Sequence $X_t \cdot \mathbb{1}(k_t)$

In Sec. D.3 we already proved that $(X_t, \mathcal{F}_t)_{t\ge1}$ is an MDS i.e., for all $t \ge 1$, $\mathbb{E}[X_t|\mathcal{F}_{t-1}] = 0$. Since $k_t$ is $\mathcal{F}_{t-1}$-measurable, we also have $\mathbb{E}[X_t\mathbb{1}(k_t)|\mathcal{F}_{t-1}] = \mathbb{1}(k_t) \cdot \mathbb{E}[X_t|\mathcal{F}_{t-1}] = 0$ with $|X_t\mathbb{1}(k_t)| \le r_{\max}D^{\mathsf{c}}$. Therefore, $(X_t\mathbb{1}(k_t), \mathcal{F}_t)_{t\ge1}$ is also an MDS. By Azuma's inequality (see for example [2, Lemma 10]):

$$\sum_{t=1}^{T}X_t\mathbb{1}(k_t) \le r_{\max}D^{\mathsf{c}}\sqrt{\frac{5}{2}T\ln\left(\frac{8T}{\delta}\right)} \quad \text{w.p.} \ge 1 - \frac{\delta}{12T^{5/4}} \tag{38}$$

## D.5 Completing the regret bound

By gathering (27), (34), (35), (36), (38) and (37) we conclude that with probability at least $1 - \frac{\delta}{12T^{5/4}}$:

$$
\begin{aligned}
\sum_{k=1}^{m} \Delta_k \cdot \mathbb{1}(k) \leq{} & 2 \left( \sqrt{28} + \sqrt{14} \right) r_{\max} \sqrt{S^{\mathsf{c}} A T \ln \left( \frac{2SAT}{\delta} \right)} \left( D^{\mathsf{c}} \sqrt{(\Gamma^{\mathsf{c}} - 1)} + 1 \right) \\
& + \frac{196}{3} r_{\max} S^{\mathsf{c}} A \ln \left( \frac{2SAT}{\delta} \right) (3 + \ln(T)) \left( D^{\mathsf{c}} S^{\mathsf{c}} + 1 \right) \\
& + 2 r_{\max} (D^{\mathsf{c}} + 1)(\sqrt{S^{\mathsf{c}} A T} + S^{\mathsf{c}} A) \\
& + r_{\max} D^{\mathsf{c}} \sqrt{\frac{5}{2} T \ln \left( \frac{8T}{\delta} \right)} + 2 \left( \sqrt{2} + 1 \right) r_{\max} \sqrt{S^{\mathsf{c}} A T} \\
& + r_{\max} D^{\mathsf{c}} \left( 1 + 2 S^{\mathsf{c}} A + S^{\mathsf{c}} A \log_2 \left( \frac{T}{SA} \right) + S^{\mathsf{c}} \right) \\
\leq{} & C \cdot \left( r_{\max} D^{\mathsf{c}} \sqrt{\Gamma^{\mathsf{c}} S^{\mathsf{c}} A T \ln \left( \frac{SAT}{\delta} \right)} + r_{\max} D^{\mathsf{c}} \left( S^{\mathsf{c}} \right)^2 A \ln^2 \left( \frac{SAT}{\delta} \right) \right)
\end{aligned}
\tag{39}
$$

where $C$ is a numerical constant independent of the MDP instance.

From (11), with probability at least $1 - \frac{\delta}{12T^{5/4}}$:

$$
\begin{aligned}
\Delta(\text{TUCRL}, T) &\leq \sum_{k=1}^{m} \Delta_k + r_{\max} \sqrt{\frac{5}{2} T \ln \left( \frac{8T}{\delta} \right)} \\
&= \underbrace{\sum_{k=1}^{m} \Delta_k \mathbb{1}(k)}_{\text{see (39)}} + \sum_{k=1}^{m} \Delta_k \cdot (1 - \mathbb{1}(k)) + r_{\max} \sqrt{\frac{5}{2} T \ln \left( \frac{8T}{\delta} \right)}
\end{aligned}
$$

where $1 - \mathbb{1}(k)$ is the complement of $\mathbb{1}(k)$ i.e., takes value 1 only when either $t_k < C(k)$ (see (13) for the definition of $C(k)$) or $M^* \notin \mathcal{M}_k$. As is proved in Appendix F.2 of [6], since both (28) and Theorem 1 of Fruit et al. [6] hold, we have that with probability at least $1 - \frac{\delta}{20T^{5/4}} \geq 1 - \frac{\delta}{12T^{5/4}}$:

$$
\sum_{k=1}^{m} \Delta_k \mathbb{1}\{M^* \notin \mathcal{M}_k\} \leq r_{\max} \sqrt{T}
\tag{40}
$$

As a consequence of (28) $t_{k+1} \leq 2t_k$. Thus, by definition of the condition $t_k < C(k)$ we have

$$
\sum_{k=1}^{m} \Delta_k \cdot \underbrace{\mathbb{1}\{t_k < C(k)\}}_{\geq 0} \leq 2 r_{\max} C(k) \leq \frac{4802}{9} r_{\max} \left( D^{\mathsf{c}} \right)^2 S^3 A \ln^2 \left( \frac{2SAT}{\delta} \right)
\tag{41}
$$

Finally, by Boole's inequality: $1 - \mathbb{1}(k) \leq \mathbb{1}\{M^* \notin \mathcal{M}_k\} + \mathbb{1}\{t_k < C(k)\}$ and so

$$
\sum_{k=1}^{m} \Delta_k \cdot (1 - \mathbb{1}(k)) \leq \underbrace{\sum_{k=1}^{m} \Delta_k \cdot \mathbb{1}\{M^* \notin \mathcal{M}_k\}}_{\text{see (40)}} + \underbrace{\sum_{k=1}^{m} \Delta_k \cdot \mathbb{1}\{t_k < C(k)\}}_{\text{see (41)}}
$$

In conclusion, there exists a numerical constant $C$ independent of the MDP instance such that for any MDP and any $T > 1$, with probability at least $1 - \frac{\delta}{12T^{5/4}} - \frac{\delta}{12T^{5/4}} - \frac{\delta}{12T^{5/4}} = 1 - \frac{\delta}{4T^{5/4}}$ we have:

$$
\Delta(\text{TUCRL}, T) \leq C \cdot \left( r_{\max} D^{\mathsf{c}} \sqrt{\Gamma S^{\mathsf{c}} A T \ln \left( \frac{SAT}{\delta} \right)} + r_{\max} \left( D^{\mathsf{c}} \right)^2 S^3 A \ln^2 \left( \frac{SAT}{\delta} \right) \right)
\tag{42}
$$

Since $\sum_{T=2}^{+\infty} \frac{\delta}{4T^{5/4}} = \delta$, by taking a union bound we have that the regret bound (42) holds with probability at least $1 - \delta$ for all $T > 1$.

**Algorithm 2** OPTIMISTIC TRANSITION PROBABILITIES (OTP) [2]

---

**Input:** Probability estimate $\widehat{p} \in \mathbb{R}^n$, confidence interval $\beta \in \mathbb{R}$, value vector $v \in \mathbb{R}^n$, subset of states $\mathcal{I} \subseteq \{s_1, \ldots, s_m\}, m \le n$, such that $\sum_{s \in \mathcal{I}} \widehat{p}(s) = 1$
**Output:** Optimistic probabilities $\widetilde{p} \in \mathbb{R}^n$

Let $\mathcal{I} = \{s_1, s_2, \ldots, s_m\}$ such that $v(s_1) \ge v(s_2) \ge \ldots \ge v(s_m)$
$\widetilde{p}_1(s_1) = \min\left\{1, \widehat{p}(s_1) + \frac{\beta}{2}\right\}$
$\widetilde{p}_1(s_j) = \widehat{p}(s_j), \quad \forall 1 < j \le m$
$j = m$
$i = 1$
**while** $\sum_{s \in \mathcal{I}} \widetilde{p}_i(s) > 1$ **do**
    $i = i + 1$
    $\widetilde{p}_i(s) = \widetilde{p}_{i-1}(s), \quad \forall s \ne s_j$
    $\widetilde{p}_i(s_j) = \max\left\{0, 1 - \sum_{s \in \mathcal{I} \setminus \{s_j\}} \widetilde{p}_{i-1}(s)\right\}$
    $j = j - 1$
**end while**
$\widetilde{p}_i(s) := 0, \ \ \forall s \in \mathcal{S} \setminus \mathcal{I}$
$\widetilde{p} := \widetilde{p}_i$

---

# E   Shortest Path Analysis

We are interesting in comparing the shortest path of any pair $(s, \overline{s}) \in \mathcal{S} \times \mathcal{S}_k^{\mathcal{C}}$ in $\mathcal{M}_k^+$ and $\overline{\mathcal{M}}_k^+$. Formally, given a target state $\overline{s}$, the stochastic shortest path $\tau_M(s) := \tau_M(s \to \overline{s})$ of an (extended) MDP $M$ is the (negation) solution of the following Bellman equation

$$
\begin{aligned}
\tau_M(s) &= -1 + \max_{a \in \mathcal{A}_s, p \in B_p(s,a)} \left\{ p^\mathsf{T} \tau_M \right\}, \qquad \forall s \ne \overline{s} \\
\tau_M(\overline{s}) &= 0
\end{aligned}
\tag{43}
$$

## E.1   Equivalence of Shortest Path in $\mathcal{M}_k^+$ and $\overline{\mathcal{M}}_k^+$

We start by proving the following.

**Lemma 6.** *For any pair* $(s, \overline{s}) \in \mathcal{S} \times \mathcal{S}_k^{\mathcal{C}}$, $\tau_{\mathcal{M}_k^+}(s \to \overline{s}) = \tau_{\overline{\mathcal{M}}_k^+}(s \to \overline{s})$.

In order to analyse the properties of the stochastic shortest path we need to investigative the maximization over the confidence interval $B_p(s, a)$ either in $\mathcal{M}_k^+$ or $\overline{\mathcal{M}}_k^+$. This problem can be solved using Alg. 2. For any state-action pair $(s, a)$, we define $\widetilde{p}_{\mathcal{M}_k^+}(\cdot|s, a) = \text{OTP}(\widehat{p}(\cdot|s, a), B_{p,k}^+(s, a), \tau, \mathcal{S})$ and $\widetilde{p}_{\overline{\mathcal{M}}_k^+}(\cdot|s, a) = \text{OTP}(\widehat{p}(\cdot|s, a), \overline{B}_{p,k}^+(s, a), \tau, \mathcal{S})$. It is easy to notice that the optimistic probability vectors built by Alg. 2 satisfy (either in $\mathcal{M}_k^+$ or in $\overline{\mathcal{M}}_k^+$)

$$\forall i \in \{1, \ldots, n\}, \qquad \widetilde{p}_i(s_1) \ge \widehat{p}(s_1)$$

$$\forall i \in \{2, \ldots, n\}, \forall l \in \{n - i + 2, n\}, \qquad \widetilde{p}_i(s_l) = \max\left\{0, 1 - \sum_{s' \ne s_l} \widetilde{p}_{i-1}(s')\right\}$$

$$= \max\left\{0, \widehat{p}(s_l) - \left(\sum_{s'} \widetilde{p}_{i-1}(s') - 1\right)\right\}$$

$$\le \widehat{p}(s_l)$$

where $s_1, \ldots, s_n$ are such that $\tau(s_1) \ge \ldots \ge \tau(s_n)$. The algorithm may stop before $n$ iterations but this means that the states not processed are kept at $\widehat{p}$.

We start considering the case in which $(s, a) \in \mathcal{K}_k$. Recall that $\forall s' \in \mathcal{S}_k^{\mathsf{T}}, \widehat{p}(s'|s, a) = 0$ by definition since $s'$ is not reachable from $\mathcal{S}_k^{\mathcal{C}}$ (i.e., $N_k(s, a, s') = 0$) and that $\mathcal{M}_k^+$ and $\overline{\mathcal{M}}_k^+$ consider

the same empirical average for the transition probabilities (i.e., $\widehat{p}$). The shortest path to $\overline{s}$ is such that $\max_s\{\tau(s)\} = \tau(\overline{s}) = 0$ and $\tau(s) \leq -1$ for any state $s \in \mathcal{S} \setminus \{\overline{s}\}$ (either in $\mathcal{M}_k^+$ or $\overline{\mathcal{M}}_k^+$). As a consequence, $s_1 = \overline{s}$ and for any $s' \in \mathcal{S}_k^{\mathsf{T}}$,

$$\widetilde{p}_{\mathcal{M}_k^+}(s') \leq \widehat{p}(s') = 0, \text{ and } \widetilde{p}_{\overline{\mathcal{M}}_k^+}(s') \leq \widehat{p}(s') = 0$$

which ensures that $\forall (s,a) \in \mathcal{K}_k$ the constraints in $\overline{\mathcal{M}}_k^+$ hold. This results is independent from the vector $v$ provided to OTP. Then, for any vector $v \in V = \{v \in \mathbb{R}^S | v(\overline{s}) = 0 \wedge v(s) \leq -1, \ \forall s \in \mathcal{S} \setminus \{\overline{s}\}\}$, we have that $\mathcal{I}^1 = \mathcal{I}^2$, since $\beta^1 = \beta^2$ and $\widetilde{p}_{\mathcal{M}_k^+}(s) = \widetilde{p}_{\overline{\mathcal{M}}_k^+}(s) = 0$ for any $s \in \mathcal{S}_k^2$ then: $\widetilde{p}_{\mathcal{M}_k^+}(s') = \widetilde{p}_{\overline{\mathcal{M}}_k^+}(s'), \ \forall s' \in \mathcal{S}$. Finally, $\forall (s,a) \in (\mathcal{S} \times \mathcal{A}) \setminus \mathcal{K}_k$ it is trivial to notice that: $\forall s' \in \mathcal{S}$, $\forall v \in V, \widetilde{p}_{\mathcal{M}_k^+}(s') = \widetilde{p}_{\overline{\mathcal{M}}_k^+}(s')$ since $B_{p,k}^+(s,a) = \overline{B}_{p,k}^+(s,a)$.

The proof follows by noticing that $\tau_{\mathcal{M}_k^+} \in V$ and $\tau_{\overline{\mathcal{M}}_k^+} \in V$.

### E.2 Bounding the bias span

**Lemma 7.** *Consider an (extended) MDP $M$ and define $L_M$ as the associated optimal (extended) Bellman operator. Given $h_0 = \mathbf{0}$, and $h_i = (L_M)^i h_0$ we have that*

$$\forall s, s' \in \mathcal{S}, \ h_i(s') - h_i(s) \leq r_{\max} \tau_M(s \to s')$$

*where $\tau_M(s \to s')$ is the minimum expected shortest path from $s$ to $s'$ in $M$.*

*Proof.* The proof follows from the application of the argument in [2, Sec. 4.3.1]. □

## F Tighter Regret Bound

In this section we present a different relaxation of $\mathcal{M}_k$ that preserves the Bernstein nature of the confidence intervals (although the final regret bound is the same). This relaxation makes the transition from TUCRL to UCRL smooth when $\mathcal{S}^{\mathsf{T}} = \emptyset$ and may perform better empirically. We initially introduced the relaxation using $\ell_1$-norm in order to prove the equivalence of the shortest paths (Lem. 6) implying that $sp_{\mathcal{S}_k^c}\{w_k\} \leq D^{\mathcal{C}}$. We now show that the same result (i.e., $sp_{\mathcal{S}_k^c}\{w_k\} \leq D^{\mathcal{C}}$) can be obtained by consider a perturbation of $B_{p,k}$ that preserves the Bernstein-like confidence intervals.

We start defining the new confidence set $\overline{Z}_p^k$ for any $(s,a,s') \in \mathcal{S} \times \mathcal{A} \times \mathcal{S}$ as

$$\overline{Z}_{p,k}(s,a,s') := \begin{cases} B_p^k(s,a,s') \text{ if } s \in \mathcal{S}_k^{\mathsf{T}} \\ B_p^k(s,a,s') \text{ if } s \in \mathcal{S}_k^{\mathsf{C}} \text{ and } p_k^+(s,a) \geq \rho_{t_k}(s,a) \\ \{0\} \text{ if } s \in \mathcal{S}_k^{\mathsf{C}}, \ p_k^+(s,a) < \rho_{t_k}(s,a), \text{ and } s' \in \mathcal{S}_k^{\mathsf{T}} \\ \left[\widehat{p}_k(s'|s,a) - \beta_{p,k}^{sas'}, \widehat{p}_k(s'|s,a) + \beta_{p,k}^{sas'} + \zeta_{p,k}^{sa}\right] \cap [0,1] \text{ otherwise} \end{cases} \tag{44}$$

where for any $(s,a) \in \mathcal{S} \times \mathcal{A}$

$$\zeta_{p,k}^{sa} := \sum_{s' \in \mathcal{S}_k^{\mathsf{T}}} p_k^+(s'|s,a) = S_k^{\mathsf{T}} \cdot \underbrace{p_k^+(s,a)}_{:=\min\left\{1, \frac{49}{3}\frac{b_{k,\delta}}{N_k^{\pm}(s,a)}\right\}} \tag{45}$$

We then define $\overline{\mathcal{M}}_k^+ := \{\mathcal{S}, \mathcal{A}, r_k(s,a) \in B_{r,k}(s,a), p_k(s'|s,a) \in \overline{Z}_{p,k}(s,a,s'), p_k(\cdot|s,a) \in \mathcal{C}\}$.

It is possible to prove that

**Lemma 8.** *For any pair $(s, \overline{s}) \in \mathcal{S} \times \mathcal{S}_k^c$, $\tau_{\overline{\mathcal{M}}_k^+}(s \to \overline{s}) \leq \tau_{\mathcal{M}_k}(s \to \overline{s})$. As a consequence, let $h_i = (L_{\overline{\mathcal{M}}_k^+})^i \mathbf{0}$, then*

$$sp_{\mathcal{S}_k^c}\{h_i\} \leq r_{\max} \max_{s,\overline{s} \in \mathcal{S}_k^c}\{\tau_{\overline{\mathcal{M}}_k^+}(s \to \overline{s})\} \leq r_{\max} \max_{s,\overline{s} \in \mathcal{S}_k^c}\{\tau_{\mathcal{M}_k}(s \to \overline{s})\} \leq r_{\max} D^{\mathcal{C}}$$

# G Proof of Lem. 3

We prove the statement by contradiction: we assume that there exists a learning algorithm denoted $\mathfrak{A}_T$ satisfying

1. for all $\varepsilon \in ]0,1]$, there exists $T_\varepsilon^\dagger \leq f(\varepsilon)$ such that $\mathbb{E}[\Delta(\mathfrak{A}_T, M_\varepsilon, x, T)] < 1/6 \cdot T$ for all $T \geq T_\varepsilon^\dagger$,

2. there exists $T_0^* < +\infty$ such that $\mathbb{E}[\Delta(\mathfrak{A}_T, M_0, x, T)] \leq C_2(\ln(T))^\beta$ for all $T \geq T_0^*$.

Any randomised strategy for choosing an action at time $t$ is equivalent to an (a priori) random choice from the set of all deterministic strategies. Thus, it is sufficient to show a contradiction when the action played by $\mathfrak{A}_T$ at any time $t$ is a deterministic function of the past trajectory $h_t := \{s_1, a_1, r_1, \ldots, s_t\}$. In the rest of the proof we assume that $\mathfrak{A}_T$ maps any sequence of observations $h_t = \{s_1, a_1, r_1, \ldots, s_t\}$ to a (single) action $a_t$.

By trivial induction it is easy to see that as long as state $y$ has not been visited, the history $h_t$ is independent of $\varepsilon$ ($\mathfrak{A}_T$ can not distinguish between different values of $\varepsilon$ and plays exactly the same action when the past history is the same).

Let's define $N_T^0(x, b) := \sum_{t=1}^{T} \mathbb{1}\{(s_t, a_t) = (x, b)\}$ the number of visits in $(x, b)$ with $a_t = \mathfrak{A}_T(h_t)$ and $\varepsilon = 0$. Note that $N_T^0(x, b)$ is not random since when $\varepsilon = 0$ both action $b$ and action $d$ loop on $x$ with probability 1. For any $\varepsilon \in [0, 1]$ and any horizon $T$ define the event:

$$F(T, \varepsilon) := \bigcap_{1 \leq t \leq T} \{s_t \neq y\}$$

where the sequence of states $s_t$ is obtained by executing $\mathfrak{A}_T$ on MDP $M_\varepsilon$. We will denote by $\overline{F(T, \varepsilon)}$ the complement of $F(T, \varepsilon)$.

For any horizon $T$, and independently of $\varepsilon$, there is only one possible trajectory $h_T = \{s_1, a_1, r_1, \ldots, s_T\}$ that never goes to $y$ and which corresponds to the trajectory observed when $\varepsilon = 0$. When $\varepsilon = 0$, the probability of this trajectory is 1 and so $\mathbb{P}(F(T, 0)) = 1$ (recall that everything is deterministic in this case) while in general we have:

$$\forall T \geq 1, \ \forall \varepsilon \in [0, 1], \ \mathbb{P}(F(T, \varepsilon)) = (1 - \varepsilon)^{N_T^0(x, b)} \tag{46}$$

We now prove by contradiction that

$$\lim_{T \to +\infty} N_T^0(x, b) = +\infty \tag{47}$$

Let's assume that $C := \max\{10, \max_{T \geq 1}\{N_T^0(x, b)\}\} < +\infty$. Taking $\varepsilon = 1/C$ and applying the law of total expectation we obtain:

$$\forall T \geq 1, \ \mathbb{E}[\Delta(\mathfrak{A}_T, M_{1/C}, x, T)] = \underbrace{\mathbb{E}\left[\Delta(\mathfrak{A}_T, M_{1/C}, x, T) | F(T, 1/C)\right]}_{=T/2 + 1/2 \cdot N_T^0(x, b) \geq T/2} \cdot \underbrace{\mathbb{P}(F(T, 1/C))}_{=(1-1/C)^{N_T^0(x, b)}}$$

$$+ \underbrace{\mathbb{E}\left[\Delta(\mathfrak{A}_T, M_{1/C}, x, T) | \overline{F(T, 1/C)}\right] \cdot \mathbb{P}\left(\overline{F(T, 1/C)}\right)}_{\geq 0}$$

$$\geq \frac{T}{2} \cdot \left(1 - \frac{1}{C}\right)^{N_T^0(x, b)} \geq \frac{T}{2} \cdot \underbrace{\left(1 - \frac{1}{C}\right)^C}_{\geq 1/3 \text{ by Lem. } 9} \geq \frac{T}{6}$$

where we used the fact that

- $N_T^0(x, b) \leq C$ and $(1 - 1/C) \in [0, 1]$ by definition, implying $\left(1 - \frac{1}{C}\right)^{N_T^0(x, b)} \leq \left(1 - \frac{1}{C}\right)^C$,
- since $C \geq 10$ we have $\left(1 - \frac{1}{C}\right)^C \geq 1/3$ by Lem. 9 applied to $x = 1/C$,
- and finally under event $F(T, 1/C)$, the regret incurred is exactly $T/2 + 1/2 \cdot N_T^0(x, b) \geq T/2$.

This contradicts our assumption that there exists $T^\dagger_{1/C} < +\infty$ such that for all $T \geq T^\dagger_{1/C}$, $\mathbb{E}[\Delta(\mathfrak{A}_T, M_{1/C}, x, T)] < T/6$ and so (47) holds.

Since $\lim_{T \to +\infty} N^0_T(x, b) = +\infty$, it is possible to construct a strictly increasing sequence $(T_n)_{n \in \mathbb{N}}$ such that:

$$\forall n \in \mathbb{N}, \ N^0_{T_{n+1}}(x, b) > N^0_{T_n}(x, b), \ T_0 = T^*_0, \ T_1 \geq C_2, \ T_1 \geq C_2 (\ln(T_1))^\beta \ \text{ and } \ N^0_{T_1}(x, b) \geq 10$$

We also define the (strictly decreasing) sequence: $\varepsilon_n := 1/N^0_{T_n}(x, b)$, $\forall n \geq 1$. By the law of total expectation:

$$\mathbb{E}[\Delta(\mathfrak{A}_{T_n}, M_{\varepsilon_n}, x, T_n)] = \underbrace{\mathbb{E}\left[\Delta(\mathfrak{A}_{T_n}, M_{\varepsilon_n}, x, T_n) | F(T_n, \varepsilon_n)\right]}_{\geq T_n/2} \cdot \underbrace{\mathbb{P}\left(F(T_n, \varepsilon_n)\right)}_{=(1-\varepsilon_n)^{N^0_{T_n}(x,b)}}$$

$$+ \underbrace{\mathbb{E}\left[\Delta(\mathfrak{A}_{T_n}, M_{\varepsilon_n}, x, T_n) | \overline{F(T_n, \varepsilon_n)}\right] \cdot \mathbb{P}\left(\overline{F(T_n, \varepsilon_n)}\right)}_{\geq 0}$$

$$\geq \frac{T_n}{2} \cdot (1 - \varepsilon_n)^{N^0_{T_n}(x,b)} = \frac{T_n}{2} \cdot \underbrace{(1 - \varepsilon_n)^{1/\varepsilon_n}}_{\geq 1/3 \text{ by Lem. 9}} \geq \frac{T_n}{6} \qquad (48)$$

where we applied Lem. 9 to $x = \varepsilon_n \leq 1/10$ since $N^0_{T_n}(x, b) \geq 10$ for all $n \geq 1$. Moreover, since by construction for all $n \geq 1$, $T_n > T_0 = T^*_0$ we have by assumption that

$$\forall n \geq 1, \ \ \mathbb{E}[\Delta(\mathfrak{A}_{T_n}, M_0, x, T_n)] = \frac{1}{2} N^0_{T_n}(x, b) = \frac{1}{2\varepsilon_n} \leq C_2 (\ln(T_n))^\beta$$

$$\implies T_n \geq \exp\left(\frac{1}{(2C_2 \cdot \varepsilon_n)^{1/\beta}}\right)$$

Since $\lim_{n \to +\infty} 1/\varepsilon_n = +\infty$ and $\lim_{x \to +\infty} \exp\left(x^{1/\beta}\right)/x^\alpha = +\infty$ there exists $N \in \mathbb{N}$ such that for all $n \geq N$, $T_n \geq f(\varepsilon_n)$. By assumption, for all $n \geq N$,

$$\mathbb{E}[\Delta(\mathfrak{A}_{T_n}, M_{\varepsilon_n}, x, T_n)] < \frac{T_n}{6}$$

which contradicts (48) therefore concluding the proof.

**Lemma 9.** *For all $x \in ]0, 1/10]$, we have $(1 - x)^{1/x} \geq 1/3$.*

*Proof.* It is easy to verify that the derivative of $x \longmapsto (1 - x)^{1/x}$ is:

$$\forall x \in ]0, 1/10], \ \ \frac{d}{dx}\left((1 - x)^{1/x}\right) = -\underbrace{\frac{(1 - x)^{1/x - 1}}{x^2}}_{\geq 0} \cdot ((1 - x)\ln(1 - x) + x)$$

It is well known that for all $x \in ]0, 1[$, $x < -\ln(1 - x) < \frac{x}{1-x}$ implying that $(1 - x)\ln(1 - x) + x$ is positive. Therefore, $\frac{d}{dx}\left((1 - x)^{1/x}\right)$ is negative on $]0, 1/10]$ implying that $x \longmapsto (1 - x)^{1/x}$ is decreasing. As a result: $\forall x \in ]0, 1/10], \ \ (1 - x)^{1/x} \geq 0.9^{10} > 1/3$. $\qquad\square$

## H  Experiments - Three-State Domain

This domain was introduced in [6] in order to show the inability of UCRL to learn in weakly communicating MDPs. The graphical representation of the domain is reported in Fig. 6. We keep the same means for the rewards (reported on Fig. 6) but we change the distributions: uniform distributions with range $1/5$ instead of Bernouillis. In the main paper we showed how the algorithms behave when $\delta = 0$. Here we consider the case the MDP is communicating by defining $\delta = 0.005$. Fig. 7 shows that, as expected, TUCRL behaves similarly to UCRL. In this example it is able to outperform UCRL since the preliminary phase in which transitions to non-observed states are forbidden leads to a more conservative exploration that, due to the structure of the problem ($s_1$ is difficult to reach but it is also non-optimal), results in a smaller regret.

Figure 6: Three-state domain introduced in [6]

Figure 7: Communicating three-state domain ($\delta = 0.005$)