[Reviews · NeurIPS 2018]

Reviewer 1



This is an excellent theoretical contribution. The analysis is quite heavy and has many subtleties. I do not have enough time to read the appended proofs; also, the subject of the paper is not in my area of research. The comments below are based on the impression I got after reading carefully the first 8 pages of the paper and glancing through the rest in the supplementary file. Summary: This paper is about reinforcement learning in weakly-communicating MDP under the average-reward criterion. The authors consider finite state-and-action MDP and the regret framework for measuring the optimality and efficiency of a learning algorithm. In the literature there are algorithms that achieve near-optimal regret bounds for communicating MDP, but none for weakly-communicating MDP (without requiring some prior knowledge about certain properties of optimal solutions of the true MDP). The authors present such an algorithm in this paper. This algorithm, called TUCRL, shares many similarities with its predecessor UCRL2, an optimistic algorithm that achieves near-optimality for communicating MDP. But it differs from UCRL2 in critical steps, in order to avoid useless attempts to reach unreachable states. Such over-exploration is indeed the major problem in applying an optimistic algorithm, like UCRL2, designed for communicating MDP to a non-communicating MDP, and it can cause a linear growth of regret. To achieve a near-optimal regret bound, the TUCRL algorithm cannot simply explore less. It also needs a careful balance between exploration and exploitation, in order to avoid giving up too easily the attempt to reach new states and thereby suffering from under-exploration. The authors prove that TUCRL meets these challenges in learning in weakly-communication MDP, and the regret bound they obtain for TUCRL is an elegant and strong theorem (Theorem 2) that parallels the results for UCRL2 (Jaksh et al. 2010). The authors also provide illuminating numerical simulations to compare the behavior of TUCRL and two other existing algorithms against what the theories predict (Section 4). Furthermore, investigating some empirical differences between TUCRL and another algorithm SCAL, the authors point out an intrinsic exploration-exploitation dilemma in learning in weakly-communicating MDP. The results for this part (Section 5) are also highly illuminating and elegant. I think this is a paper of exceptionally high quality. It provides a satisfactory answer to the open question on learning in weakly-communicating MDP, for the case when the agent starts from the communicating part of the MDP. It also clarifies and corrects several erroneous claims in the literature (Appendices A-B). Regarding presentation, it is very challenging to explain the heavy analysis and numerous (important) subtleties involved in this work and to discuss the meaning of the results at a high level. The authors have achieved that. I find their writing almost impeccable. Other fairly minor comments: 1. About the abstract: -- line 4: It seems by "This leads to defining weakly-communicating ... MDP," the authors mean "This results in weakly-communicating ... MDP" instead. -- line 9 (and also line 36): The phrase "the longest shortest path" is an oxymoron. To explain the diameter $D^C$, a full phrase (e.g., "the longest path among the shortest paths between state pairs") would be better. -- line 10: It seems better to change "in contrast with optimistic algorithms" to "in contract with existing optimistic algorithms," since the authors' algorithm is also of the optimistic type. 2. line 56, p.2: A reference or explanation is needed for "the doubling trick." 3. It would be good to explain what the acronym TUCRL stands for. Maybe add "(TUCRL)" after the title of Section 3? 4. Typos in p.3: -- In Eq. (2), the dot in $\hat{p}(\cdot | s, a)$ should be $s'$. -- In the displayed equations between line 125 and 126, comma is misplaced inside the second equation. 5. Section 3: When explaining the algorithm, it would be good to mention also that by construction, the extended MDP given to the EVI algorithm is always weakly-communicating, so the EVI algorithm always terminates in finite time. 6. line 178-179, p. 5: It would be good to provide a reference for the fact mentioned in this sentence. 7. TEVI algorithm, top of p. 14: Should the set $B_p(s,a)$ be $\bar{B}^+_p(s,a)$ instead?

Reviewer 2



Paper presents an algorithm, TUCRL for efficient exploration-exploitation balance in weakly communicating MDPs. It is claimed that for weakly communicating MDPs no algorithm can achieve logarithm regret without first encountering linear regret for a time that is exponential in the parameters of the MDP. Authors also point out a mistake in the proof of REGAL [3]. The results of this paper are exciting, however, it is a difficult paper to parse. There are many symbols and they are not centrally defined which makes reading this paper an unpleasant experience. I would suggest the authors include a table/chart that the reader can refer to know what different symbols stand for. Some other comments: (a) The abstract starts off by saying it is not possible to achieve some states in Mountain car -- it will be good to a give a few examples of it in the text/supplementary materials. (b) Line 302-303: a proof of Lemma 6 is provided in supp. materials, the use of language such as "seems straightforward to adapt", "SCAL probably satisfies 2" is unwarranted. Either SCAL satisfies or it doesnot -- not sure why authors use the langauge that makes the reader belief that SCAL maynot satisfy 2 etc. Please revise the language and be precise. (c) It would be good to see the results of the proposed method on more tasks such as mountain car, that the authors allude to. I am not an expert in finding bounds for RL algorithms, but the results look exciting and proofs appear to be correct (I did not verify all lines of all proofs). The reason I have given this paper is a lower rating is because it is not easy to read the paper.

Reviewer 3



UPDATE: I have read the authors' response and it was very helpful and addressed my concerns well and I have adjusted my score. I now see the flaw in the regal reasoning and encourage the authors to add the explicit callout of the backwards inequality in the supplementary material (unless I missed it). I also agree with the reasons for not comparing to regal empirically. The point about algorithm mechanics is well thought out, though I still see some benefit to calling it out. The wording correction you mentioned for line 159 makes sense as well. Summary: The paper considers the problem of RL exploration/exploitation in weakly communicating MDPs where many states (S_T) may not actually be reachable. Such states cause an overestimate of values in algorithms like UCRL, which consider such states in their contribution to the exploration bonus. The authors propose a new algorithm, TUCRL, that performs an online estimate of the reachability of states, and discards states that are below a threshold based on visits and the number of steps so far. The algorithm is shown to have improved regret in such MDPs. Furthermore, a new lower bound is proven showing that no algorithm in this setting can achieve logarithmic regret without first incurring exponential regret. Review: Overall, the paper makes a strong case for TUCRL. The proofs are thorough and convincing. I appreciate having the sketches actually in the paper in addition to the complete results in the supplementary material. The idea appears to be sound, since UCRL was not making use of the tighter bounds that can be derived on state reachability, and the empirical results verify that the real-world advantages of the improved bound are significant. The lower bound is also an important result outside of the TUCRL analysis. My only concerns are in the comparison to Regal, some of the terminology in Section 2, and the description of the algorithm mechanics but I think these can be easily addressed. The paper’s claim of a flaw in the proof of Regal’s regret was surprising and the logic laid out in the supplementary material appears correct, but the explanation is worded in a confusing way that makes it unclear what exactly the flaw is. On line 417 of the supplemental material, we see the flawed inequality -C \Sum{v} <= -C 2^j. The authors then state: “[that inequality] is not true since -C := -c / sqrt(2^j). However, that definition of C does not actually invalidate the inequality since C appears on both sides. The authors later point out that what is needed is a lower bound on v, but they are unable to find one so they declare the proof incorrect. All the evidence here is pointing to there being a flaw and the authors being right, but we’re missing an actual example where the inequality fails to hold. The absence of a proof does not mean the statement is necessarily wrong (even though I believe it is). Can the authors give us an actual MDP or even just plausible numerical instantiations of this inequality that make it untrue? If so, that would seem to be a much cleaner refute of Regal. In addition to the unclear terminology above, I was surprised that Regal did not appear in the experiments since it seems to be built for these kinds of environments, even if the theory is flawed. Would an empirical result actually show TUCRL outperforming Regal? If all we have is questions about its proof then an empirical result seems needed. Most of the paper’s terminology is rigorously defined, but I found the definition of S^C and S^T in the beginning of Section 2 surprisingly informal. The initial definitions make it unclear what cases are allowed under these conditions – are they hard partitions where only one is visited – can there be many “island” MDPs that don’t communicate but an episode can start in any one of them? In that case, what counts as S^C? Many of these questions are cleaned up laterin the text, particularly in the multi-chain MDP subsection later in section 2, but they should really be addressed with a formal and rigorous definition of S^C and S^T right at the start. One of the most interesting parts of the proof was the bounding of the number of times a poorly visited (s,a) is visited. This is reminiscent of escape probabilities in the PAC literature but in TUCRL, we see a UCB/UCRL like behavior where “jumps” in the exploration occur periodically (though with longer and longer periods). This is also borne out in the empirical results. I would have liked to have seen this intuition presented when the algorithm was introduced, maybe as a set of “key properties” of the algorithm so that users will better understand the behavioral characteristics of the algorithm. Right now those details are scattered in various sections. I realize space is tight, but you could add a few lines to the supplemental material covering this. The main result is a good mathematical derivation but the significance of the empirical results put the theory in a better light. The lower bound is also a strong contribution and an important component for a theory paper. Minor notes: Line 159: poorly visited so far are enabled -> This is unclear. What does enabled mean? Shouldn’t poorly visited states be discarded?